**Night-time oxidation of surfactants at the air–water interface: effects of chain length, head group and saturation.**

Federica Sebastiani,[a,b] Richard A. Campbell,[b] Kunal Rastogi[a] and Christian Pfrang.[a*]

[a] *Department of Chemistry, University of Reading, P.O. Box 224, RG6 6AD, Reading, UK*
[b] *Institut Laue-Langevin, 71 avenue des Martyrs, CS20156, 38042 Grenoble Cedex 9, France*
* corresponding author: c.pfrang@reading.ac.uk

**Abstract**

Reactions of the key atmospheric night-time oxidant $NO_3$ with organic monolayers at the air–water interface are used as proxies for the ageing of organic-coated aqueous aerosols. The surfactant molecules chosen for this study are oleic acid (OA), palmitoleic acid (POA), methyl oleate (MO) and stearic acid (SA) to investigate the effects of chain length, head group and degree of unsaturation on the reaction kinetics and products formed. Fully and partially deuterated surfactants were studied using neutron reflectometry (NR) to determine the reaction kinetics of organic monolayers with $NO_3$ at the air–water interface for the first time. Kinetic modelling allowed us to determine the rate coefficients for the oxidation of OA, POA and MO monolayers to be $(2.8 \pm 0.7) \times 10^{-8}$ cm$^2$ molecule$^{-1}$ s$^{-1}$, $(2.4 \pm 0.5) \times 10^{-8}$ cm$^2$ molecule$^{-1}$ s$^{-1}$ and $(3.3 \pm 0.6) \times 10^{-8}$ cm$^2$ molecule$^{-1}$ s$^{-1}$ for fitted initial desorption lifetimes of $NO_3$ at the closely packed organic monolayers, $\tau_{d,NO3,1}$, of $8.1 \pm 4.0$ ns, $16 \pm 4.0$ ns and $8.1 \pm 3.0$ ns, respectively. The approximately doubled desorption lifetime found in the best fit for POA compared to OA & MO is consistent with a more accessible double bond associated with the shorter alkyl chain of POA facilitating initial $NO_3$ attack at the double bond in a closely packed monolayer. The corresponding uptake coefficients for OA, POA and MO were found to be $(2.1 \pm 0.5) \times 10^{-3}$, $(1.7 \pm 0.3) \times 10^{-3}$ and $(2.1 \pm 0.4) \times 10^{-3}$, respectively. For the much slower $NO_3$-initiated oxidation of the saturated surfactant SA we estimated a loss rate of approximately $(5 \pm 1) \times 10^{-12}$ cm$^2$ molecule$^{-1}$ s$^{-1}$ which we consider to be an upper limit for the reactive loss, and estimated an uptake coefficient of ca. $(5 \pm 1) \times 10^{-7}$. Our investigations demonstrate that $NO_3$ will contribute substantially to the processing of unsaturated surfactants at the air–water interface during night-time given its reactivity is ca. two orders of magnitude higher than that of $O_3$. Furthermore, the relative contributions of $NO_3$ and $O_3$ to the oxidative losses vary massively between species that are closely related in structure: $NO_3$ reacts ca. 400 times faster than $O_3$ with the common model surfactant oleic acid, but only ca. 60 times faster with its methyl ester MO. It is therefore necessary to perform a case-by-case assessment of the relative contributions of the different degradation routes for any specific surfactant. The overall impact of $NO_3$ on the fate of saturated surfactants is slightly less clear given the lack of prior kinetic data for comparison, but $NO_3$ is likely to contribute significantly to the loss of saturated species and dominate their loss during night-time. The retention of the organic character at the air–water interface differs fundamentally between the different surfactant species: the fatty acids studied (OA and POA) form products with a yield of ~ 20% that are stable at the interface while $NO_3$-initiated oxidation of the methyl ester MO rapidly and effectively removes the organic character ($\leq$ 3% surface-active products). The film-forming potential of reaction products in real aerosol is thus likely to depend on the relative proportions of saturated and unsaturated surfactants as well as the head group properties. Atmospheric lifetimes of unsaturated species are much longer than those determined with respect to their reactions at the air–water interface, so that they must be protected from oxidative attack *e.g.* by incorporation into a complex aerosol matrix or in mixed surface films with yet unexplored kinetic behaviour.

*Keywords:* aerosol surface, atmospheric reactions, oleic acid, palmitoleic acid, methyl oleate, stearic acid, nitrate radicals, neutron reflectometry.

**1. Introduction**

Over the last decades, aerosols have attracted increasing attention from the scientific community because their impact on the Earth's radiative balance and on cloud formation is still largely unknown (Shindell et al., 2009; Stevens et al., 2009; Stocker et al., 2013). Atmospheric aerosols derive from natural processes (e.g. volcanoes, wind-blown dust and sea-spray) and from human activities (e.g. combustion and cooking). A key feature for the aerosol behaviour is the presence of organic material both in the bulk and at the surface (Fuzzi et al., 2006). Organic compounds contained in atmospheric aerosols are often surface-active, such as fatty acids. Atmospheric fatty acids include saturated (such as palmitic acid; Adams & Allen, 2013) as well as unsaturated acids *e.g.* oleic acid which is found as component of marine (Tervahattu et al., 2002a; Tervahattu et al., 2002b; Fu et al., 2013) and cooking (Allan et al., 2010) aerosol. Cooking emissions have been estimated to contribute ca. 10% to the man-made emission of small particulate matter ($PM_{2.5}$) at 320 mg per person per day based on measurements in London (Ots et al., 2016). The composition and lifetime of aerosol particles in the atmosphere are largely determined by the ageing process due to exposure to trace gases, such as $NO_3$, OH, $O_3$ or other oxidants (e.g. Cl and Br; Estillore et al., 2016). To study the aerosol ageing it is crucial to investigate the heterogeneous reactions occurring between the particles and gas-phase oxidants. While homogeneous chemistry is well described at the molecular level, the study of heterogeneous reactions remains a major challenge. Field measurements suggest that heterogeneous reactions may change the chemical composition of particles and in particular of their surface films (Robinson et al., 2006). The reactions may alter important properties of the particles like aerosol hydrophilicity, toxicity and optical properties. Most of the studies to date have investigated the heterogeneous reaction of organic aerosols by $O_3$ and OH, which are the main oxidants during daytime. During night-time, [OH] is very low while the concentration of the photo-labile $NO_3$ will build up and becomes significant. Therefore while OH controls the chemistry of the daytime atmosphere, $NO_3$ radicals have a similar role during the night (Wayne et al., 1991; Mora-Diez et al., 2002; Ng et al., 2017). In many cases heterogeneous reactions have been studied using organic droplets or thick films (e.g. King et al., 2004; Gross et al., 2009). However, it has been shown that experimental studies of organic molecules self-assembled at the surface of water rather than purely organic aerosols alone are key to understanding atmospheric ageing of aerosols covered in organic material (Vesna et al., 2008).

In the work presented here organic monolayers at the air–water interface are used as proxies for the organic-coated aqueous atmospheric aerosols, and their reactions with $NO_3$ are investigated. The molecules chosen for this study are oleic acid (OA), palmitoleic acid (POA), methyl oleate (MO) and stearic acid (SA). OA (King et al., 2004; King et al., 2009; King et al., 2010), POA (Huff Hartz et al., 2007; Pfrang et al., 2011), MO (Hearn et al., 2005; Zahardis & Petrucci, 2007; Xiao & Bertram, 2011; Pfrang et al., 2014, Sebastiani et al., 2015) and SA (Sobanska et al., 2015) are popular model systems for atmospheric surfactants. MO, the methyl ester of OA, is a main component of biodiesel (chemical name: fatty acid methyl esters or 'FAME'; Wang et al., 2009) likely leading to an increased atmospheric abundance in the future since up to 7% of FAME is added to standard petroleum diesel in the EU to reduce greenhouse gas emissions; higher proportions of FAME in petroleum diesel

(10% FAME sold as 'B10' and 20% FAME sold as 'B20') as well as pure FAME ('B100') become increasingly common fuel alternatives across a number of European countries including Germany, France and Finland.

This selection of molecules allows the investigation of the effects of chain length, head group and degree of unsaturation on the reaction kinetics and products formed. The surface excess of the organic molecule during the oxidation reaction is monitored using neutron reflectometry (NR). NR is a powerful technique that can be used to determine the surface excess of a deuterated monolayers at the air–ACMW (air contrast matched water) interface (Lu et al., 2000), and information about reaction mechanisms can even be accessed thanks to partial deuteration of the surfactant (Thompson et al., 2010; Thompson et al., 2013). Further, the surface composition of mixed systems can be resolved in situ during dynamic processes by the selective deuteration of different components (Campbell et al., 2016; Ciumac et al., 2017), and therefore the reaction rates of individual components in mixtures holds great potential for future studies. In the present work, NR is used effectively to measure the surface excess of organic material (i.e. the combination of reactants and insoluble, involatile products) in situ during reactions with gas-phase $NO_3$.

The study of heterogeneous reactions of $NO_3$ at the air–water interface is made possible thanks to four recent key advances. First, the high flux and the stability of the neutron reflectometer FIGARO (Fluid Interfaces Grazing Angles ReflectOmeter, Campbell et al., 2011) at the Institut Laue-Langevin (Grenoble, France) is exploited through the acquisition of data at the air–water interface that is far faster than was previously possible (King et al., 2009; King et al., 2010). Second, surface excesses down to monolayer coverage on the order of a few percent can now be determined precisely through a refined method of background treatment (Pfrang et al., 2014). Third, improvements in the sample environment have been achieved by the design and commissioning of a new reaction chamber that has a gas delivery system optimised for homogeneous diffusion (Sebastiani et al., 2015). Lastly, rigorous measurements of the oxidant concentrations and development of a kinetic model (Pöschl et al., 2007; Shiraiwa et al., 2009; Shiraiwa et al., 2010) to interpret the data have been undertaken. Specifically, $NO_3$ is produced in situ by reacting $O_3$ with $NO_2$, the dependence of $[NO_3]$ on the initial $[NO_2]$ and $[O_3]$ is modelled, and to determine the concentration of $NO_3$, the steady state concentrations of $NO_2$ and $N_2O_5$ are measured using FTIR spectroscopy as a function of the initial $[NO_2]$.

The analysis of the kinetic experiments required the development of a modelling approach to describe all the relevant reactions and processes. In order to describe the $NO_3$-initiated oxidation we used a model, which considers, in addition to reactions, other mechanisms, such as accommodation, desorption, competition for adsorption sites and transport of the gas-phase species. This model builds on the formalism and terminology of the "PRA framework" (introduced by Pöschl, Rudich and Ammann in Pöschl et al., 2007). It is a combination of K2–SURF , kinetic double-layer surface model (Shiraiwa et al., 2009) and KM–SUB, kinetic multi-layer model of aerosol surface and bulk chemistry (Shiraiwa et al., 2010), but has been adapted to a planar geometry. KM–SUB and K2–SURF have been applied to describe a range of experimental datasets and conditions (e.g. Pfrang et al., 2011). Both models describe the evolution of the kinetic parameters of an organic droplet exposed to oxidants. We have adapted the model to a monomolecular organic layer at the air–water interface for analysis and interpretation of the experimental data presented here. The kinetic analysis of the measured surface excess

decays for the four reaction systems provides information on the rate coefficients of the heterogeneous reaction
as well as indirect information on the formation of surface-active products. The results obtained for the different
molecules will be discussed in relation of their chemical structures. Furthermore, the comparison between $NO_3$
and other oxidants species indicates to what extent night-time oxidation is important to atmospheric aerosol
ageing. We also estimated oxidant uptake coefficients and compared those to literature data on similar organic
molecules that have been studied in the condensed phase (i.e. droplets or thick films; King et al., 2004 and Gross
et al., 2009).
**2. Methods**
**2.1. Experimental**
*2.1.1 Materials*
The organic monolayers comprised either deuterated oleic acid ($d_{34}$OA, $CD_3(CD_2)_7CD=CD(CD_2)_7CO_2D$, Sigma-
Aldrich, isotopic purity ≥ 98%, purity 99%), partially deuterated palmitoleic acid ($d_{14}$POA,
$CH_3(CH_2)_5CH=CH(CD_2)_7CO_2H$, custom-synthesised by the Oxford Deuteration Facility), deuterated methyl
oleate ($d_{33}$MO, $CD_3(CD_2)_7CD=CD(CD_2)_7CO_2CH_3$, custom-synthesised by the Oxford Deuteration Facility, ~
95%) and deuterated stearic acid ($d_{35}$SA, $CD_3(CD_2)_{16}CO_2H$, Sigma-Aldrich, isotopic purity 98%, purity 99%);
further details may be found in section 1 of the ESI; the chemical structures of the molecules studied are
displayed in Scheme 1. The subphase was a mixture of 8.1% by volume $D_2O$ (Sigma Aldrich) in pure $H_2O$
(generated using a Millipore purification unit, 18.2 MΩ cm), known as air contrast matched water (ACMW).
Chloroform (Sigma-Aldrich, > 99.8%) and $O_2$ (Air Liquide, France, > 99.9%) were used as supplied. $NO_2$ was
supplied in small gas cylinders (112 dm$^3$) by Scientific and Technical Gases Ltd (Newcastle-under-Lyme, UK)
and provided as a mixture with synthetic air at a concentration of 1000 ppm with an analytical tolerance of ± 2%.
The solutions of organic molecules in chloroform were prepared shortly before the experiments and the
concentrations are given in mg of solute in volume of solution: for $d_{34}$OA 1.41 mg ml$^{-1}$, for $d_{14}$POA 1.26 mg
ml$^{-1}$, for $d_{33}$MO 1.11 mg ml$^{-1}$ and for $d_{35}$SA 0.58 mg ml$^{-1}$.

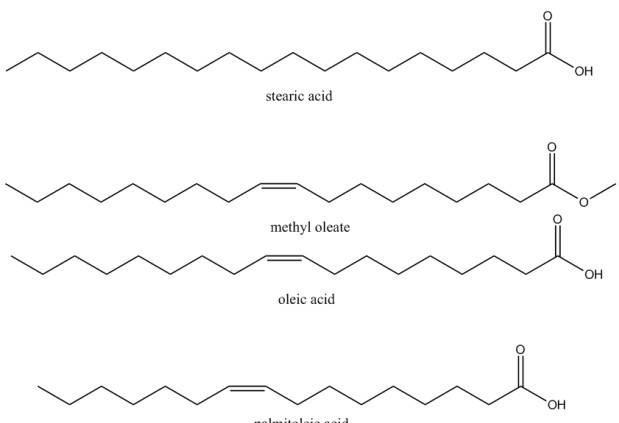

**Scheme 1.** Chemical structures of the organic molecules studied.
*2.1.2 Gas Delivery*
Nitrate radicals, $NO_3$, were produced *in situ* from the reaction of $O_3$ with $NO_2$. $O_3$ was generated by the exposure
of molecular oxygen to UV light (the procedure has been described elsewhere; Pfrang et al., 2014). $[NO_3]$ was
regulated by changing the flow rate of $NO_2$ in the range $0.045 - 0.23$ $dm^3$ $min^{-1}$ while $[O_3]$ was kept constant at
3.9 ppm (i.e. using a constant UV exposure of the $O_2$ molecules and a fixed $O_2$ flow rate of 1.2 $dm^3$ $min^{-1}$). A
flow of the $NO_3$-$NO_2$-$N_2O_5$-$O_2$ mixture was then admitted to the reaction chamber (Sebastiani et al., 2015) and
the organic monolayer was oxidised at a rate that was determined by $[NO_3]$; we ensured that the reaction
chamber as well as the reaction bulb where $NO_2$ was allowed to react with $O_3$ to form $NO_3$ was kept in the dark
to avoid any photolysis of the photolabile $NO_3$ during the experiments. Measurements of $NO_2$ and $N_2O_5$ were
carried out using IR absorption spectroscopy to establish the concentrations, $[NO_2]$ and $[N_2O_5]$, and their
uncertainties. Modelling of the well-known reaction scheme allowed the estimation of $[NO_3]$. At a total flow rate
of l.2 to 1.5 $dm^3$ $min^{-1}$, $[NO_3]$ ranged from $(3.5 \pm 1.5) \times 10^8$ ($13 \pm 5$ ppt) to $(2.3 \pm 1.2) \times 10^9$ molecule $cm^{-3}$ (86
$\pm$ 45 ppt) in the experiments presented here; $[NO_3]$ and $NO_2$ flow rates are given in Table 1. From the gas
reaction model it is found that $NO_2$ reaches the steady state concentration faster when initial $[NO_2]$ is higher.
Ozone is consumed quantitatively in less than 250 s (see Fig. 7 in Section 3.1 of ESI). The concentration of $NO_3$
is lower the higher the excess of $NO_2$ (see Fig. 9 in Section 3.1 of ESI). The steady state concentrations of $N_2O_5$
are always approaching a similar value (see Fig. 8 in Section 3.1 of ESI) that is determined by the initial ozone
concentration.
**Table 1.** The concentrations of $NO_3$ calculated from IR measurements of $[NO_2]$ and $[N_2O_5]$ are reported in
the first column as molecule $cm^{-3}$ and the corresponding ppt value is given in the second column; in the
third column the flow rate of $NO_2$ is shown (the total gas mixture flow rate is obtained by adding the
constant $O_2$ flow rate of 1.2 $dm^3$ $min^{-1}$ to these values).

| $NO_3$ / molecule $cm^{-3}$ | $NO_3$ / ppt | $NO_2$ flow rate / $dm^3$ $min^{-1}$ |
|---|---|---|
| $(3.5 \pm 1.5) \times 10^8$ | $(13 \pm 5)$ | 0.360 |
| $(4.2 \pm 1.4) \times 10^8$ | $(15 \pm 5)$ | 0.290 |
| $(6.1 \pm 1.2) \times 10^8$ | $(23 \pm 4)$ | 0.200 |
| $(9 \pm 3) \times 10^8$ | $(32 \pm 10)$ | 0.160 |
| $(10 \pm 3) \times 10^8$ | $(36 \pm 10)$ | 0.130 |
| $(9.3 \pm 2.4) \times 10^8$ | $(35 \pm 9)$ | 0.104 |
| $(2.3 \pm 1.2) \times 10^9$ | $(86 \pm 45)$ | 0.08 |

The modelled concentrations were confirmed by IR measurements of $[NO_2]$ and $[N_2O_5]$ (the full dataset is
displayed in Section 3.2 of ESI). Further details on the gas flow system as well as the $NO_3$ modelling may be
found in Sections 2 and 3 of the ESI.
*2.1.3 Neutron Reflectometry (NR)*
NR measurements of the oxidation of deuterated monolayers by $NO_3$ in the reaction chamber (Sebastiani et al.,
2015) were carried out on FIGARO at the Institut Laue-Langevin (Campbell et al., 2011). High flux settings
were used to maximise the data acquisition rate involving an incident angle, $\vartheta$, of 0.62°, a wavelength, $\lambda$, range
of $2 - 20$ Å, and a constant resolution in momentum transfer, $q$, of 11% over the probed $q$-range of 0.007 to 0.07
$Å^{-1}$, where $q = 4\pi \sin \vartheta / \lambda$.
Only a brief description of the physical basis of NR with reference to its application is given while more details
may be found in Lu et al. (2000), Narayanan et al. (2017) and Braun et al. (2017). NR is a technique that can be

used to measure the surface excess of oil-like films at the air–water interface. The scattering of neutrons is related to the coherent cross sections of the atoms with which they interact, and these values vary non-monotonically with respect to different isotopes of the same atom and different atoms across the periodic table. In particular, swapping hydrogen for deuterium in molecules changes significantly the scattering, and as such mixing of hydrogenous and deuterated materials enables contrast matching.

The time-of-flight mode allowed us to follow the change in reflectivity of a deuterated monolayer at the air–water interface simultaneously over the whole measured $q$-range with respect to the time of the oxidation reaction. For a deuterated surfactant monolayer at the air–ACMW interface the reflectivity, $R$, can be expressed by:

$$R \cong \frac{16\pi^2}{q^4} 4b^2 n^2 \sin^2\left(\frac{qd}{2}\right) \tag{1}$$

where $b$ is the scattering length of the surfactant, in fm, $n$ is the number density, in $\text{Å}^{-3}$, $d$ is the thickness of the layer in Å, and $bn = \rho$ is the scattering length density. The surface excess, $\Gamma$, is given by:

$$\Gamma = \frac{1}{A_{hg}} = \frac{\rho d}{b} \tag{2}$$

where $A_{hg}$ is the area per molecule (or per head group). The surface excess for insoluble molecules corresponds to the surface concentration.

A stratified layer model was applied to the experimental data involving a single layer for the deuterated surfactant. It has been shown that in such a case and in this low $q$-range (< 0.07 $\text{Å}^{-1}$), the value of $\Gamma$ is very insensitive to specific details of the model applied (Angus-Smyth et al. 2012). Therefore, fitting of the thickness with an arbitrary fixed value of the density or fitting of the density with an arbitrary fixed value of the thickness (each within reasonable bounds) gives equivalent results to within an added uncertainty of < 2 %. That is, only the fitted product $\rho d$ directly determines $\Gamma$, and the measurement approach deliberately desensitizes the data to structural information such as the actual layer thickness during the reaction in order to gain the requisite kinetic resolution. In our case, we chose to fit $\rho$ while fixing $d$ at the value obtained by fitting data recorded over a wider $q$-range (up to 0.25 $\text{Å}^{-1}$).

Normalisation of the reflectivity data was carried out with respect to the total reflection of an air–$D_2O$ measurement. The sample stage was equipped with passive and active anti-vibration control. The reaction chamber was mounted on the sample stage, it was interfaced with the gas setup, and the trough was filled with 80 ml of ACMW. A given amount of solution was spread using a microlitre syringe in order to form the monolayer following the protocol use in other NR studies of atmospheric relevance (Pfrang et al. 2014; Sebastiani et al. 2015; Skoda et al. 2017; King et al. 2010; Thompson et al. 2010; King et al. 2009). The volume of solution spread was 24 µl for $d_{34}$OA, 23 µl for $d_{14}$POA, 32 µl for $d_{33}$MO and 35 µl for $d_{35}$SA. The solvent was allowed to evaporate before closing the chamber. The trough in the reaction chamber did not have barriers to compress the film and adjust the surface pressure, hence the desired surface pressure, in the range of 16 to 25 mN m$^{-1}$ depending on the molecule, was achieved by spreading a calculated number of molecules on the water surface. Off-line tests using a surface pressure sensor confirmed that the surface pressure could be achieved reproducibly – between 2 to 7 % variation depending on the molecule – and the stability of the assembled film was assessed for 3–4 hours by monitoring the surface pressure or the reflectivity profile. From the surface excess obtained by NR the reproducibility is found to be within 1 to 9 %, depending on the molecule. The choice of

initial surface pressure and surface excess was based on the requirement of maximising the signal-to-noise ratio
for NR measurements while having a reaction that lasts long enough to be analysed for kinetics parameters. A
reduction of the initial surface pressure is not expected to affect the kinetic behaviour, i.e. the $\Gamma(t)$ will start from
a lower value and the curve will extend on a shorter time and less data will be available for the kinetic fitting. An
increase of spread molecules will produce more droplets floating on top of a monolayer, when the molecule is
unsaturated (compare to Figures 1–3 in Section 1 of ESI), while it will introduce inhomogeneity in the
monolayer formed by saturated molecule (see Fig. 4 Section 1 of ESI) preventing a reliable interpretation of the
NR measurement. The monolayer was further characterised with compression-expansion isotherms with a
Langmuir trough off-line, while recording Brewster-angle microscopy (BAM) images at different surface
pressure values, and these results are shown in the ESI Section 1. Data were recorded for a few minutes before
$NO_3$ was admitted into the chamber. The time resolution was 2 s. The alignment of the interface was maintained
to a precision of 5 μm using an optical sensor (LK-G152, Keyence, Japan; laser class II, wavelength 650 nm,
power output 0.95 mW, spot diameter 120 μm), which operated through the laser alignment window of the
reaction chamber (Sebastiani et al., 2015).
**2.2. Kinetic modelling**
Oxidation of organic compounds by $NO_3$ may proceed via several reaction channels: rapid addition to the double
bond of unsaturated species as well as slower abstraction of hydrogen atoms particularly relevant for saturated
compounds (Wayne et al., 1991). These mechanisms as well as transport processes need to be considered in
order to fit our experimental data. Based on the PRA-framework (Pöschl et al., 2007; Shiraiwa et al., 2009;
Shiraiwa et al., 2010; Pfrang et al., 2010; Shiraiwa et al., 2012a), a specific model has been developed for the
heterogeneous reaction of a monomolecular organic layer at the air–water interface. The oxidant loss due to the
reaction and transport to the bulk water has been taken into account. The organic reactants used in the
experiments show a very low solubility and slow diffusion in water, hence the loss due to transport to the bulk
could be neglected. The product branching ratios of the heterogeneous reactions are not known, and we were not
able to identify individual product compounds from a monomolecular film at the air–water interface. The
products were thus divided into three categories: volatile, soluble and surface-active species. The distinction
between soluble and volatile species is made on the basis of the product yields reported previously (Hung et al.,
2005; Docherty & Ziemann, 2006) for bulk reaction and considering vapour pressures (Compernolle et al., 2011)
and solubilities (Kuhne et al., 1995) of the products. In the model, the branching ratios for volatile and soluble
products are based on literature values, and for surface-active products an estimation was based on $\Gamma(t)$ at long
reaction times; the technique used in this study monitors the deuterium concentration at the interface, no other
information can be obtained. We could have described the reaction system by assuming only two types of
products: surface active and non-surface active. However, we decided to distinguish non-surface active
compounds between volatile and soluble products in order to make our model suitable for description of
experimental data probing the partitioning to subphase and/or gas-phase. Because of the method used to produce
$NO_3$ (see Section 2.1.2 and Sections 2–3 of ESI) the ratio $[NO_2]/[NO_3]$ increases from $10^5$ to $10^7$ as $[NO_3]$
decreases from $10^9$ to $10^8$ molecule $cm^{-3}$. Since $NO_2$ can adsorb and desorb from the organic layer (compare
King et al., 2010), occupying reactive sites for an average time represented by the desorption lifetime, the loss of
organic material due to reaction with $NO_3$ may also be affected. The $NO_2$ occupies a reactive site, which
becomes unavailable for $NO_3$ oxidation, and hence reduces the number of reactive sites available and slows
down the apparent reaction rate. In particular, for high $[NO_2]/[NO_3]$ ratios the reactant loss rate will be lower
than the loss rate recorded for the lower $[NO_2]/[NO_3]$ ratios. To take this effect into account we included the
absorption and desorption of $NO_2$ in the model and to describe it we introduced the parameter called desorption
lifetime, $\tau_{d,NO2}$, following the approach used by Shiraiwa et al. (2009). The effect of $N_2O_5$ is not considered in
the model, since the concentration was constant for all gas conditions, as shown in Figure 8 of the ESI.
Experimental studies of reactive uptakes of $NO_3$ and $N_2O_5$ (Gross & Bertram, 2008; Zhang et al., 2014a;
Gržinic, et al., 2015) have shown that $NO_3$ uptake is substantially faster with a comparative study for OA
reporting a ca. four orders of magnitude higher uptake coefficient of $NO_3$ compared to $N_2O_5$ (Gross et al., 2009).
The system has been modelled as a gas phase (g) and a near-surface gas phase (gs), above a sorption layer (s), a
surface layer (ss), a near-surface bulk (nb) and the bulk (b), following the formalism of Shiraiwa et al. (2010) (as
illustrated in Figure 1). Different to the model presented by Shiraiwa et al. (2010) we had to remove the
curvature terms from the modelling code to be able to describe the flat air–water interface present in our
experimental system. We do not expect any significant impact of curvature on the processes studied here.

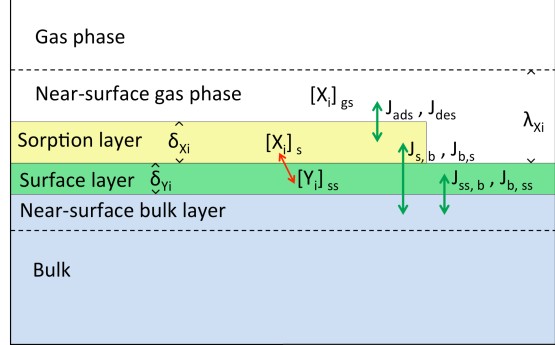

**Figure 1.** Kinetic model for an organic layer at the air–water interface, $\delta_{Xi}$ and $\delta_{Yi}$ are the thicknesses of
sorption and surface layer. $\lambda_{Xi}$ is the mean free path of $X_i$ in the gas phase. The red arrow shows chemical
reactions. The green arrows show the transport fluxes.
The gas-phase species, $NO_3$ and $NO_2$, can adsorb to the sorption layer and interact with the organic molecules in
the surface layer. The products can stay at the surface layer, or they can be lost through solubilisation into the
bulk or by evaporation into the gas phase.
The evolution of the gas species surface concentration, $[X_i]_s$, can be described by taking into account the
following processes: adsorption, desorption, transport and reaction. Full details are given in the ESI. In the
following section, only the key equations that describe the reactions are discussed (the nomenclature used is
based on the PRA framework; Pöschl et al., 2007; Shiraiwa et al., 2009; Shiraiwa et al., 2010; Pfrang et al.,
2010; Shiraiwa et al., 2012a).
Our gas-phase species $NO_3$ reacts with the organic layer and the loss, $L_{surf,Y,NO3}$, can be described with the
second–order rate coefficient $k_{surf,Y,NO3}$:
$$L_{surf,Y,NO_3} = k_{surf,Y,NO_3}\,[Y]_{SS}\,[NO_3]_S \qquad (3)$$
The evolution of the $NO_3$ surface and bulk concentrations can be described as follows:
$$\frac{d[NO_3]_s}{dt} = J_{ads,\,NO_3} - J_{des,\,NO_3} - L_{surf,Y,NO_3} + J_{bs,\,NO_3} - J_{sb,\,NO_3} \qquad (4)$$
$$\frac{d[NO_3]_b}{dt} = \left(J_{sb,NO_3} - J_{bs,NO_3}\right)\frac{A}{V} \tag{5}$$
where $A$ is the water surface area and $V$ is the total water volume. In the case of $NO_2$ the corresponding equation
4 does not have the $L_{surf}$ term, since it is not reactive toward the organic molecules considered (King et al. 2010),
Eq. 5 is the same. The flux of adsorbed gas molecules, $J_{ads,\,NO_3}$, is proportional to the surface accommodation
coefficient, $\alpha_{s,NO3}$, is determined by the product of the surface accommodation coefficient on an adsorbate–free
surface, $\alpha_{s,0,NO3}$, and the sorption layer coverage $\theta_s$ which is given by the sum of the surface coverage of all
competing adsorbate species (see details in Section 4.1 of ESI). The flux of desorption, $J_{des,\,NO_3}$, is proportional
to the inverse of the desorption lifetime, $\tau_{d,\,NO_3,eff}^{-1}$, which is the average time that the $NO_3$ molecule occupies an
adsorption site. $\tau_{d,\,NO_3,eff}^{-1}$ is a combination of two desorption lifetimes, depending on the organic molecule
packing at the interface, $\theta_{ss} = [Y]_{ss}(t)/[Y]_{ss}(0)$; either closely packed ($\tau_{d,\,NO_3,1}^{-1}$), or in the gas-like state
($\tau_{d,\,NO_3,2}^{-1}$):
$$J_{des,\,NO_3} = k_{d,NO_3}[NO_3]_s = \tau_{d,\,NO_3,eff}^{-1}[NO_3]_s \tag{6}$$
$$\tau_{d,\,NO_3,eff}^{-1} = \theta_{ss}\tau_{d,\,NO_3,1}^{-1} + (1 - \theta_{ss})\tau_{d,\,NO_3,2}^{-1} \tag{7}$$
The organic reactant, Y, (e.g. oleic acid) can be lost just through reaction with $NO_3$ at the surface, hence it is
described as:
$$\frac{d[Y]_{ss}}{dt} = -k_{surf,Y,NO_3}[Y]_{SS}[NO_3]_S \tag{8}$$
The products (Z) of the heterogeneous reaction cannot be identified individually at the air–water interface by the
experimental techniques used, hence we divided them in three main categories: surface-active (i.e. remaining at
the surface and directly measurable by NR, $Z_S$), volatile (i.e. escaping into the gas-phase, $Z_G$) and soluble (i.e.
accumulating the droplet bulk, $Z_B$) species. Since the surface-active products ($Z_S$) will remain at the air–water
interface, the surface–bulk transport is neglected:
$$\frac{d[Z_S]_{ss}}{dt} = c_S k_{surf,Y,NO_3}[Y]_{SS}[NO_3]_S \tag{9}$$
where $c_S$ is the branching ratio for the surface-active products. The volatile products ($Z_G$) will leave the surface
depending on their vapour pressures, but with a lack of information on the chemical composition, we decided to
use a first-order loss rate coefficient, $k_{loss,G}$, to describe the overall effect, hence the differential equation for $Z_G$
is:
$$\frac{d[Z_G]_{ss}}{dt} = c_G k_{surf,Y,NO_3}[Y]_{SS}[NO_3]_S - k_{loss,G}[Z_G]_{ss} \tag{10}$$
where $c_G$ is the branching ratio relative to the volatile products. The bulk–surface transport is not considered for
the volatile products because it is assumed to be negligible compared to the volatilisation process. The soluble
products ($Z_B$), once formed, will diffuse into the water bulk depending on the diffusion coefficient, $D_{b,B}$, and the
transport velocity can be estimated as $k_{bss,B} \approx 4 D_{b,B}/\pi\delta_B$, where $\delta_B$ is the effective molecular diameter of the
soluble species. The inverse process is described by a surface–bulk transport velocity $k_{ssb,B} \approx k_{bss,B}/\delta_B$, hence
the evolution of the soluble product concentration in surface layer (ss) and bulk (b) is expressed as:

1. $$\frac{d[Z_B]_{ss}}{dt} = c_B k_{\text{surf,Y,NO}_3} [Y]_{SS} [NO_3]_S + k_{\text{bss,B}}[Z_B]_b - k_{\text{ssb,B}}[Z_B]_{ss} \qquad (11)$$

2. $$\frac{d[Z_B]_b}{dt} = \left(k_{\text{ssb,B}}[Z_B]_{ss} - k_{\text{bss,B}}[Z_B]_b\right)\frac{A}{V} \qquad (12)$$

3. where $c_B$ is the branching ratio for the soluble products. The equations (4)–(12) describe the evolution of the

4. various species. This system of equations cannot be solved analytically, hence the ODE solver of MATLAB

5. (2011) has been used for numeric solving. In order to fit $\Gamma(t)$, provided by NR, a minimisation of the value of $\chi^2$

6. has been performed using the FMINUIT package (Allodi).

7. The product branching ratios affect the whole $\Gamma(t)$, varying $c_S$ the final value of $\Gamma(t)$ changes, i.e. a higher $c_S$

8. leads to a higher final value of $\Gamma(t)$; the model is less sensitive to changes in $c_G$ and $c_B$, however change in the

9. solubilisation and/or volatilisation kinetic parameters ($D_{b,B}$ and $k_{\text{loss,G}}$) will affect the decay of $\Gamma(t)$. These

10. parameters were chosen in order to best describe the experimental data and taking into account literature data.

11. The kinetic model described above depends on several parameters, and some of them are strongly correlated.

12. For example, for a given gas species time evolution, which may be described by certain accommodation

13. coefficients ($\alpha_{s,0,Xi}$ where $X_i$ is $NO_3$ or $NO_2$) and certain desorption lifetimes ($\tau_{d,Xi}$), a good fit may be obtained

14. as well with a lower $\alpha_{s,0,Xi}$ combined with a higher $\tau_{d,Xi}$. The accommodation coefficient represents the

15. probability of the gas-phase molecule to absorb at the organic layer, hence the lower $\alpha_{s,0,NO3}$ is, the smaller is the

16. probability of the reaction with the organic molecule. The desorption lifetime represents the mean residence time

17. of the molecule absorbed at the surface, hence the longer this time, the higher is the probability for the gas

18. molecule to react (valid for $NO_3$). $NO_2$ does not react with the organic layer (King et al. 2010), but those

19. parameters still compensate, because $\alpha_{s,0,NO2}$ determines the number of molecules absorbed and $\tau_{d,NO2}$ determines

20. the number of molecules leaving the sorption sites. The choice of leaving both of these parameters free to vary in

21. the fitting will lead to a wide range of values for both. The resulting surface excess will match the experimental

22. data. However, the choice of fixing one out of these two parameters makes the optimisation of the model

23. computationally easier and the comparison between different organic molecules possible. In the fitting we have

24. fixed the $\alpha_{s,0,Xi}$ to one for both gas species.

25. The desorption lifetime for the reactive species, $NO_3$, shows a correlation to the reaction rate coefficient,

26. $k_{\text{surf,Y,NO}_3}$, for example if the rate coefficient is kept constant an increase in desorption lifetime will lead to

27. higher loss rate, and vice versa, if $\tau_{d,NO_3,\text{eff}}$ is kept constant and $k_{\text{surf,Y,NO}_3}$ increases the loss rate will augment.

28. Our measurement follows the loss rate, the values for $k_{\text{surf,Y,NO}_3}$ and $\tau_{d,NO_3,\text{eff}}$ are obtained from the best fit of

29. the model to the data.

30.

31. **3. Results**

32. Three of the organic molecules considered in this work (OA, POA and MO) contain one unsaturated C=C bond

33. in the aliphatic tail while one molecule (SA) is fully saturated. Among the unsaturated surfactants, POA has a

34. shorter tail than OA and MO, whereas MO is a methyl ester in comparison with the fatty acids OA and POA.

35. The double bond is expected to be the key reactive site for $NO_3$. Kinetic data on the three reactive unsaturated

36. surfactants are presented first in Sections 3.1 to 3.3, respectively. Furthermore, in a separate process $NO_3$ is

37. known to abstract hydrogen atoms from the aliphatic tail of organic molecules (Shastri & Huie, 1990; Wayne et

al., 1991; Mora-Diez et al., 2002). In order to investigate this effect as well, kinetic data on the saturated surfactant is then presented in Section 3.4.

### 3.1. Oleic acid ($d_{34}$OA) exposed to nitrate radicals (NO$_3$)

Figure 2 shows the surface excess decays of $d_{34}$OA monolayers at the air–ACMW interface as a function of time with respect to [NO$_3$]. The NO$_3$-initiated oxidation leads to a non-zero surface excess value (7–10 × 10$^{13}$ molecule cm$^{-2}$) at the end of the reaction. This plateau value is reached after an initial decay, which lasts between 5 min and over 1 h depending on [NO$_3$]. [NO$_3$] ranges from (13 ± 6) to (86 ± 45) ppt. For several gas conditions, the oxidation was carried out twice, demonstrating a good reproducibility for high [NO$_3$] (> 35 ppt), and higher variability for lower concentrations. However, the uncertainty in [NO$_3$], for [NO$_3$] < 35 ppt, is ~ 30%, which means that even a small variation in concentration produces a measurable change in the rate of loss of material. For example, such an effect can explain the differences of the $d_{34}$OA loss rates recorded for [NO$_3$] = 15 ppt. The oxidant flows in the chamber at $t = 0$ s, but the decays of the surface excess show a delayed loss most clearly seen at low [NO$_3$] (black traces with [NO$_3$] = 13 ppt). The duration of this initial plateau is longer when the oxidant concentration is lower. This suggests that some lenses of oleic acid may be floating on top of the monolayer, and they act as a reservoir for the monolayer until they are totally consumed, then the decay visible by NR relates only to the monolayer. Brewster angle microscopy (BAM) images, recorded while the OA monolayer was compressed, show the appearance of lenses, which are not visible in the expanded phase (see Section 1 of ESI). The surface excess of $d_{34}$OA was monitored as well for exposure to O$_2$ and NO$_2$ in order to assess a mechanical loss due to gas flux and isomerisation effects due to the presence of NO$_2$ (King et al., 2010).

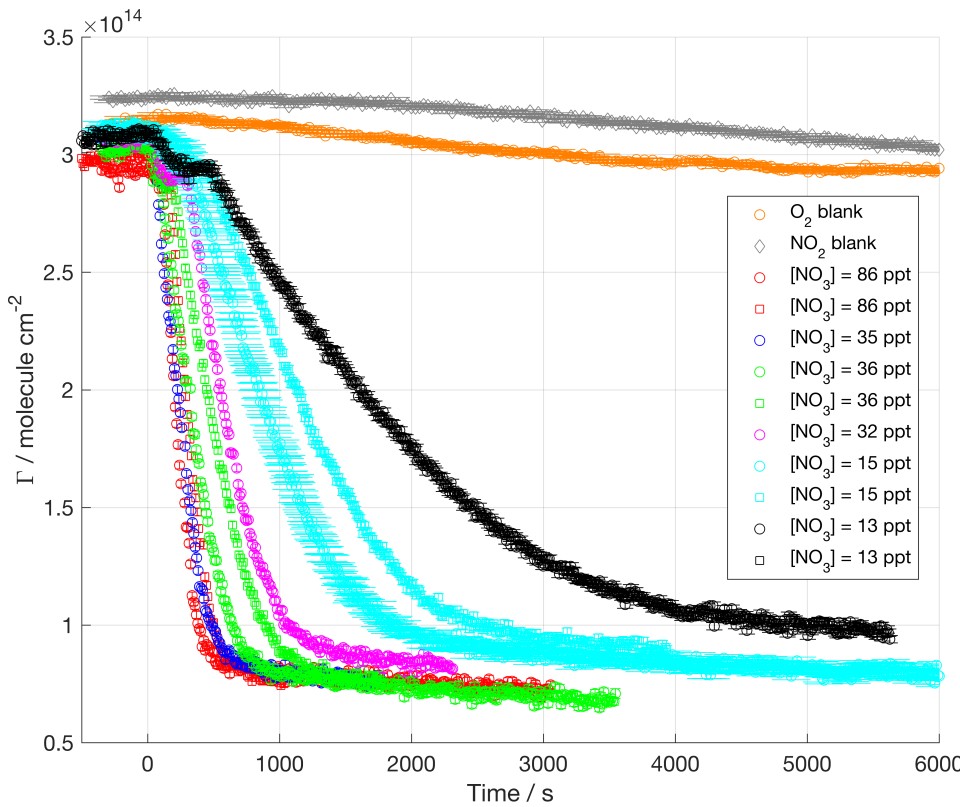

**Figure 2.** Surface excess decays of oleic acid ($d_{34}$OA) exposed to different [NO$_3$]; mean values of NO$_3$ mixing ratios are displayed in the legend (1 ppt = 2.7 × 10$^7$ molecule cm$^{-3}$). NO$_3$ is admitted at $t = 0$ s.

The kinetic fitting was performed taking into account the variability of the gas concentrations (both for $NO_3$ and
$NO_2$) and the initial surface excess was set to a suitable value to take into account the presence of oleic acid
droplets and their contribution to products. An example of the kinetic fit is displayed in Figure 3 (see ESI for the
complete data set).

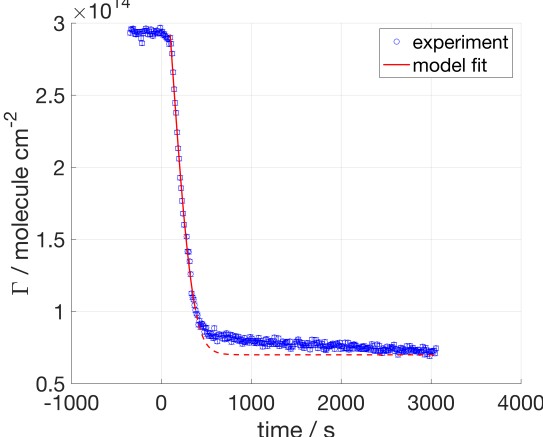

**Figure 3.** Oleic acid ($d_{34}$OA) exposed to $[NO_3]$ = 86 ppt. The red line illustrates the fit obtained from our kinetic
modelling. The solid section of the red line indicates the data range used for the optimisation of the kinetic
parameters; the dashed section of this line illustrates the modelled final part of the decay, but these data were not
used in the optimisation of the fitting since below a certain surface excess the molecules rearrange with a
different orientation in respect to the interface. The experimental data are displayed with error bars but they are
of the same scale as the marker size and hence not very visible; these experimental uncertainties were used in the
fitting procedure to calculate the value of $\chi^2$.
The range of data used for the kinetic fitting starts after the initial plateau, and ends at $1 \times 10^{14}$ molecule cm$^{-2}$:
data below this value are excluded from the fitting for two main reasons: (i) at low coverage the data become
more sensitive to experimental details such as the precise background subtraction, so the parameters that affect
the kinetic model are better determined without increasing sensitivity to these factors; and (ii) at low coverage
some surfactants can segregate into domains which are inhomogeneous laterally, and the NR model does not
have the resolution to distinguish this effect but the results are modestly affected, so again it is better to
desensitize the kinetic parameters to this effect. The fitted curve, which results from the sum of the surface
excesses of $d_{34}$OA and the products, is shown as a solid red line in Figure 3. Since NR effectively measures the
quantity of deuterium atoms at the air–ACMW interface, a distinction between reactant and products is not
possible; hence the fitting function needs to take into account the contribution to $\Gamma$ from both $d_{34}$OA and its
reaction products. In order to determine the product yields, it is assumed that at $t$ = 0 s the signal is arising solely
from $d_{34}$OA, while the signal for long reaction time (e.g. $t$ > 1000 s for $[NO_3]$ = 86 ppt) is entirely due to the
surface-active products. Also, the products (Hung et al., 2005; Docherty & Ziemann, 2006) are assumed to have
a similar scattering length density to $d_{34}$OA, on the basis that upon oxidation the $d_{34}$OA molecule is expected to
break into two parts (Hung et al., 2005; Docherty & Ziemann, 2006), which each maintains almost the same
ratio between scattering length and molecular volume. In a first approximation, the scattering length of the
products is likely to be half of the scattering length of $d_{34}$OA and the product film thickness can be thought to be
ca. half of the $d_{34}$OA film thickness. Given that and considering Eq. 2, the resulting surface excess of the
products corresponds to the value calculated with $\rho$, $d$ and $b$ of $d_{34}$OA. This approximation is not valid in the
extreme case of the products being only surface-active, since the packing would be two times denser than that
for oleic acid, and this should be considered in the surface excess calculation and consequent modelling. In our
study, the surface-active product yield is 20% and it has been taken into account that the total number of product
molecules (surface-active, volatile and soluble) was twice the number of the reactant molecules; we have also
estimated the scattering length densities for the likely products.
The accommodation coefficients for the gas-phase species were fixed to one, and the desorption lifetimes were
left free to vary in the range $10^{-9}$–$10^{-7}$ s, which is in agreement with the values suggested by Shiraiwa et al.
(2012b). For the rate coefficient, $k_{surf}$, the range of variability was optimised through a preliminary sensitivity
study performed by changing in the Matlab code the value of $k_{surf}$. The suitable range of values found was (0.7–
4) × $10^{-8}$ cm$^2$ molecule$^{-1}$ s$^{-1}$, which is significantly higher than the best fit value provided by Shiraiwa et al.
(2012b) for abietic acid exposed to NO$_3$ (1.5 × $10^{-9}$ cm$^2$ molecule$^{-1}$ s$^{-1}$). The optimisation of the kinetic
parameters was performed systematically by the $\chi^2$ minimisation routine FMINUIT (Allodi). Modelled
evolutions of the concentrations of reactants and products are exemplified in Figure 4.

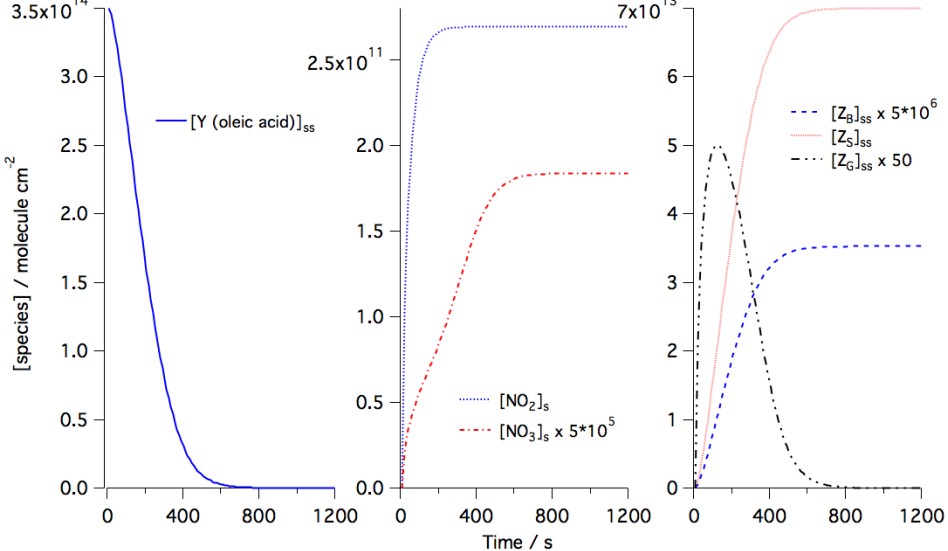

**Figure 4.** The evolution of the surface concentrations obtained from kinetic modelling using the best-fitted
parameters for the data shown in Fig. 3 for (a) the organic reactant (Y) in this case oleic acid; (b) the gas-phase
species NO$_3$ and NO$_2$; and (c) the surface-active (Z$_S$), volatile (Z$_G$) and soluble (Z$_B$) products.
From BAM images (see ESI) we know that droplets of organic molecules float on top of the monolayer, we need
to account for this extra molecules when fitting the model to the experimental data. In fact, the molecules of the
monolayer and the droplets are consumed upon oxidation but, until droplets are present, they act as a reservoir
and further molecules from the droplets may spread and maintain a constant surface excess until the droplets
disappear, leading to the delayed start in decay. The NR signal is averaged over a large surface (cm$^2$) and it is
not sensitive to small droplets (μm) thicker than the monolayer, that is why the surface excess value is constant
for this initial part of the decay. To account for this, the initial value for the theoretical $\Gamma(t)$ was adjusted to a
higher value than the initial experimental plateau value and the experimental data were considered for fitting
after the initial plateau ended (see Figure 5). Figure 5 displays a sensitivity study that demonstrates how the
change of desorption lifetimes can affect the model while keeping all the other parameters to the best fit
values. A decrease of $\tau_{d,\,NO_3,2}$ slows down the loss rate, especially for the second half of the decay, while an
increase of $\tau_{\mathrm{d,NO_3,1}}$ speeds up the decay substantially. A decrease in $\tau_{\mathrm{d,NO2}}$ does not affect the model
significantly ($\tau_{\mathrm{d,NO2}}$ was reduced by four orders of magnitude to see any effect in Fig. 5), while an increase slows
down the loss rate. Fig. 5 illustrates that the rate coefficients derived through modelling should be quoted
together with the desorption lifetimes obtained for the best fit given the substantial impact of changes in the
desorption times on the fit to the experimentally observed decays.

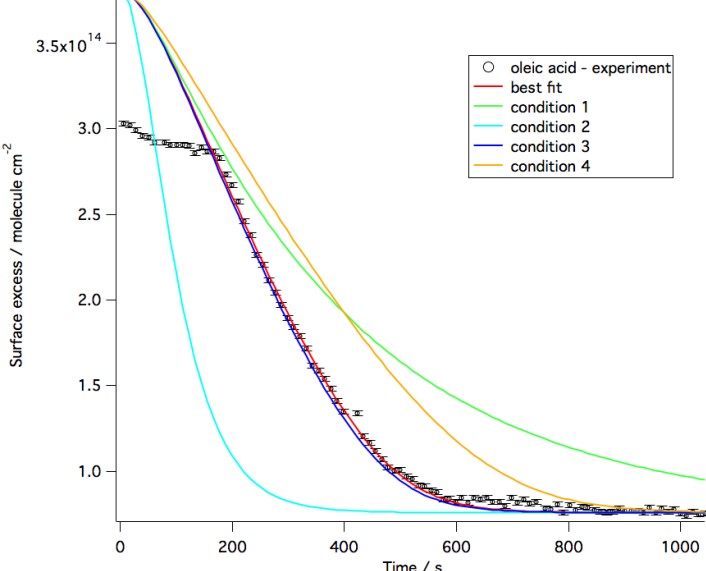

**Figure 5.** The experimental data for dOA exposed to [NO₃] = 86 ppt are shown with the best fit in red. The
desorption lifetimes for NO₃ and NO₂ have selectively been modified in this sensitivity study to show their
effect on the modelled surface excess decay. Condition 1 refers to $\left(\tau_{\mathrm{d,NO_3,1}}\right)_{best\,fit} = \tau_{\mathrm{d,NO_3,2}}$, and hence
$\tau_{\mathrm{d,NO_3,2}} = \frac{1}{6}\left(\tau_{\mathrm{d,NO_3,2}}\right)_{best\,fit}$. Condition 2 refers to $\tau_{\mathrm{d,NO_3,1}} = \left(\tau_{\mathrm{d,NO_3,2}}\right)_{best\,fit}$ and hence $\tau_{\mathrm{d,NO_3,1}} =$
$6\left(\tau_{\mathrm{d,NO_3,1}}\right)_{best\,fit}$. Condition 3 refers to $\tau_{\mathrm{d,NO2}}=10^{-4}\,(\tau_{\mathrm{d,NO2}})_{best\,fit}$. Condition 4 refers to $\tau_{\mathrm{d,NO2}}=15\,(\tau_{\mathrm{d,NO2}})_{best\,fit}$.
This fitting approach has been applied to all the molecules studied, while accounting for different product yields
and kinetic parameter ranges (see Table 2).
A preliminary analysis of the $\Gamma(t)$ profiles was needed to choose the kinetic parameters related to the products,
which have been used as fixed input parameters. The product yields were optimised to $c_S = 0.2$ for the surface-
active products, $c_G = 0.45$ for the volatile products and $c_B = 0.35$ for the soluble products. The product yields
were derived from Docherty & Ziemann (2006); the products were assumed to be hydroxy nitrates, carbonyl
nitrates, dinitrates and hydroxydinitrates (Docherty & Ziemann, 2006) as well as a dimer and more highly
nitrated compounds from Hung et al. (see products 2a' and 2b' in Hung et al., 2005). A systematic study was
performed to determine the effect of the loss of volatile and soluble products on the resulting surface excess
profiles. For the volatile products, it was found that a first-order loss rate coefficient, $k_{\mathrm{loss,G}}$, above $1 \times 10^{-1}\,\mathrm{s}^{-1}$
does not change the $\Gamma(t)$ profile and a value of $5 \times 10^{-1}\,\mathrm{s}^{-1}$ was chosen. For the soluble products, the loss will
occur upon diffusion in the sub-phase, hence the relevant parameter is the diffusion coefficient into the bulk
water, $D_{\mathrm{b,ZB}}$. The calculated $\Gamma(t)$ was affected by the presence of soluble products only for values of $D_{\mathrm{b,ZB}}$ below
$10^{-14}\,\mathrm{cm}^2\,\mathrm{s}^{-1}$; since no evidence of such an effect was found in the experimental data $D_{\mathrm{b,ZB}}$ was fixed to $10^{-7}$
$cm^2 \ s^{-1}$. The best fit values for the kinetic parameters related to the heterogeneous reaction between $d_{34}OA$ and
$NO_3$ are summarised in Table 2. The rate coefficient for $d_{34}OA$–$NO_3$ reaction in presence of $NO_2$ and $O_2$ is (2.8
$\pm$ 0.7) $\times$ $10^{-8}$ $cm^2$ $molecule^{-1}$ $s^{-1}$. The loss due to $O_2$ and/or $NO_2$ flows leads to an apparent rate coefficient on
the order of $10^{-11}$ $cm^2$ $molecule^{-1}$ $s^{-1}$, which is well within the uncertainty of the reactive rate coefficient. The
short desorption time obtained for the best fit for $NO_3$ is (8.1 $\pm$ 4.0) $\times$ $10^{-9}$ s and the slow desorption is about
three times longer, similar to the $NO_2$ desorption time.  The introduction of two desorption times reflects the
change of orientation of the organic molecules at the interface, i.e. for a highly packed monolayer the reactive
site is assumed to be less accessible, and the oxidant has less affinity for other parts of the molecules hence the
desorption is faster. When the organic surface coverage decreases the reactive sites become more accessible and
the desorption is slowed down. The effect of the two desorption time on the $[NO_3]_s$ evolution is visible in Figure
4, where the increase of $[NO_3]_s$ shows a different slope from 200 s once the oleic acid surface excess halved
(compare to Eq. 7). Figure 4 shows the time evolution of the surface concentrations of reactants, products and
gas-phase species; once the reactant, $d_{34}OA$, is completely consumed all the other species reach a steady state.
**Table 2.** Results of the kinetic modelling of the experimental data for the $d_{34}OA$–$NO_3$, $d_{14}POA$–$NO_3$, $d_{33}MO$–
$NO_3$ and $d_{35}SA$–$NO_3$ systems.  The uncertainties correspond to one standard deviation.

| | **Best fit values** | | | |
|---|---|---|---|---|
| **Modelled parameter** | $d_{34}OA$ | $d_{14}POA$ | $d_{33}MO$ | $d_{35}SA$ |
| $k_{surf}$ / $10^8$ $cm^2$ $molecule^{-1}$ $s^{-1}$ | 2.8 $\pm$ 0.7 | 2.4 $\pm$ 0.5 | 3.3 $\pm$ 0.6 | (5 $\pm$ 1) $\times$ $10^{-4}$ |
| *(constraints)* | (0.7 – 4) | (1 – 3) | (0.7 – 4) | ($10^{-4}$ – 4) |
| $\tau_{d,NO3,1}$ / $10^9$ s | 8.1 $\pm$ 4.0 | 16 $\pm$ 4.0 | 8.1 $\pm$ 3.0 | 18.2 $\pm$ 0.4 |
| *(constraints)* | (5 – 20) | (5 – 20) | (5 – 20) | (5 – 20) |
| $\tau_{d,NO3,2}$ / $10^8$ s | 2.3 $\pm$ 0.8 | 3.1 $\pm$ 1.3 | 3.7 $\pm$ 1.3 | $[0.70 \pm 0.01]^a$ |
| *(constraints)* | (0.7 – 4) | (1 – 6) | (1 – 5) | (0.7 – 4) |
| $\tau_{d,NO2}$ / $10^8$ | 2.8 $\pm$ 1.6 | 4.7 $\pm$ 2.0 | 2.9 $\pm$ 2.0 | 4.7 $\pm$ 0.4 |
| *(constraints)* | (0.1 – 6) | (0.1 – 6) | (0.1 – 6) | (0.1 – 6) |

[a] $\tau_{d,NO3,2}$ corresponds to the lower limit of the constrained range; in this system the surface excess does
not halve in the experimentally accessible timeframe and hence $\tau_{d,NO3,2}$ is not accurately determined.
**3.2. Palmitoleic acid ($d_{14}POA$) exposed to nitrate radicals ($NO_3$)**
$NO_3$-initiated oxidation of POA monolayers at the air–water interface was studied as described above for OA. 14
deuterium atoms were present between the carbon double bond and the carboxylic group in the partially-
deuterated $d_{14}POA$ sample used. POA has a chemical structure that is similar to OA. In fact the portion from the
carboxylic acid to the C=C bond is exactly the same, while the remaining part of POA chain has just five $CH_2$
units compared to the seven $CH_2$ units present in the corresponding part of the OA chain. The key reactive site
(C=C) for $NO_3$-initiated oxidation is in a similar chemical environment, but the products formed and their fates
may be different. Products are expected to be analogous to those formed by oleic acid, except that they should be
slightly more volatile since the alkyl chain is shorter.
Figure 6 shows the surface excess decays of $d_{14}POA$ monolayers at the air–ACMW interface as a function of
time with respect to $[NO_3]$. The reaction leads to a non-zero surface excess in the range 3 – 7 $\times$ $10^{13}$ molecule
$cm^{-2}$, which is slightly lower than the value found for $d_{34}OA$; this suggests that a proportion of the surface-active
products is formed of hydrogenous material and hence has a low scattering contrast to the neutron probe. The
proportion of molecules remaining stably at the interface in relation to the number of initial reactant molecules is
15% for $d_{14}$POA while it is 20 to 25% for $d_{34}$OA (depending on which initial surface excess value is used, fitted
or measured). On the assumption that the double bond is the reactive site and breaks during the oxidation
process, the partial deuteration of the $d_{14}$POA (as opposed to the full deuteration of $d_{34}$OA) may in fact help in
determining which part of the molecule remains at the interface: 5–10% of the surface-active products appear to
originate from the alkyl chain not connected to the acidic head group in the $d_{34}$OA system (however, a direct
proof would require for half-deuterated $d_{34}$OA and/or fully deuterated $d_{14}$POA to become available for additional
oxidation experiments).
For low oxidant concentrations ([NO$_3$] < 32 ppt), the final plateau value was not always reached (although it was
reached for the slowest reaction) because the reaction had to be stopped prematurely due to time constraints of
beam time experiments. Compared to $d_{34}$OA, the decay signals are more noisy, which is due to the half
deuteration leading to a weaker contrast and hence lower signal to noise ratios. The decays of surface excess
start as soon as NO$_3$ is admitted to the chamber and no initial plateau is visible (as was the case for some of the
$d_{34}$OA decays displayed in Figure 2). No lenses were formed in this system, as was confirmed by recording
BAM images while the POA monolayer was compressed (see ESI).

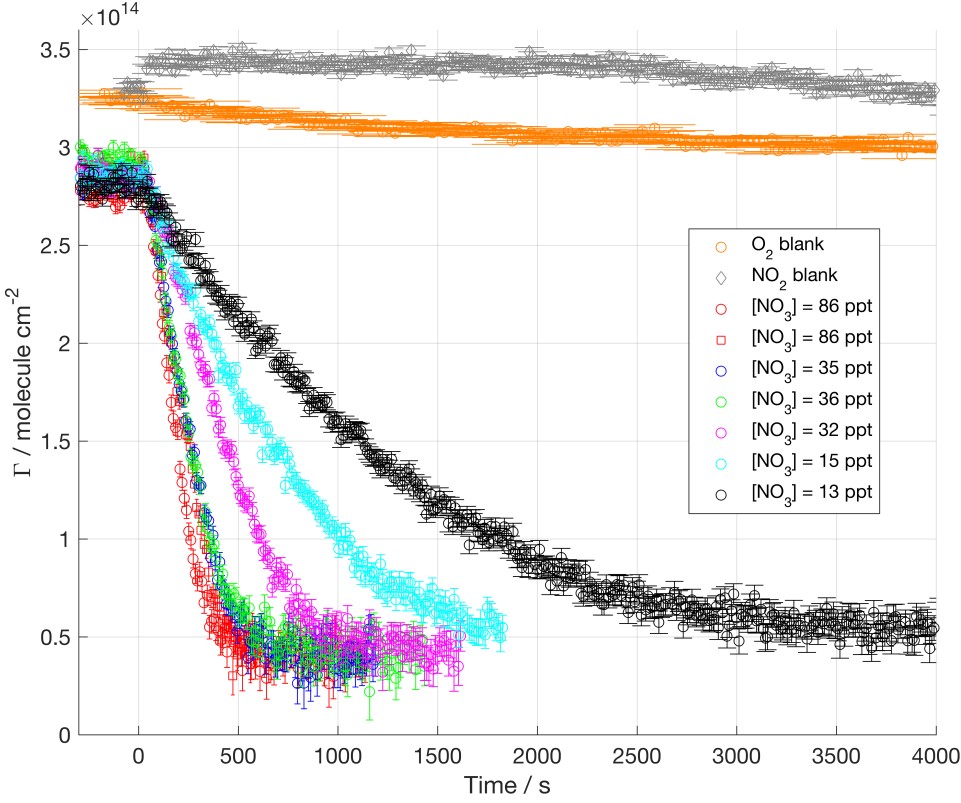

**Figure 6.** Surface excess decays of palmitoleic acid ($d_{14}$POA) exposed to different [NO$_3$]; mean values are
displayed in the legend. NO$_3$ exposure is started at $t$ = 0 s. The experimental data are more scattered than those
for $d_{34}$OA, because the $d_{14}$POA was half-deuterated (*i.e.* 14 D atoms, see Table 1 in ESI) leading to a weaker
contrast (*i.e.* lower signal-to-noise ratio) compared to the fully deuterated molecules studied.
The kinetic analysis was performed as described for $d_{34}$OA. The input parameters for description of the products
were $c_S$ = 0.17, $c_G$ = 0.48 and $c_B$ = 0.35, the surface-active and volatile product yields were adjusted to match the
residual surface excess; please note that hydrogenous surface-active products are not taken into account in this
context since the experimentally observed signal originates exclusively from the deuterated part of the POA
molecules. The variable parameters were constrained to the following value ranges: $k_{surf}$ was allowed to vary $(1 -$
$3) \times 10^{-8}$ cm$^2$ molecule$^{-1}$ s$^{-1}$, $\tau_{d,NO3,1}$ varied between $(5 - 20) \times 10^{-9}$ s, $\tau_{d,NO3,2}$ varied between $(10 - 60) \times 10^{-9}$ s
and $\tau_{d,NO2}$ varied between $(0.1 - 6) \times 10^{-8}$ s (see Table 2).

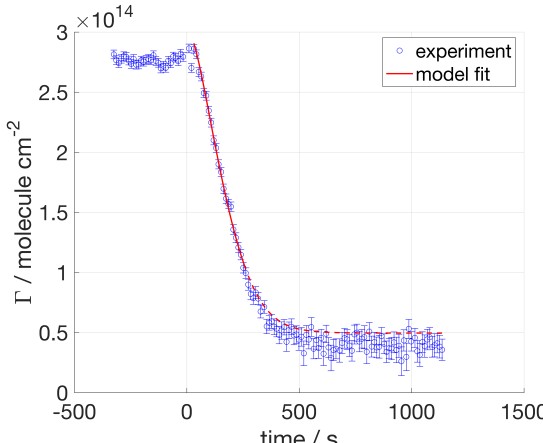

**Figure 7.** Palmitoleic acid ($d_{14}$POA) exposed to [NO$_3$] = 86 ppt. The red line illustrates the fit obtained from our
kinetic modelling (the solid section of the line indicates the data range used for the kinetic analysis; the dashed
section of the model line illustrates the calculated final part of the decay, but the corresponding experimental
data were not used in the optimisation of the fitting).
In Figure 7 an example of the model fitted to $d_{14}$POA data is displayed; the decay is very well represented by the
model. The results of the kinetic modelling for $d_{14}$POA are presented in Table 2. While the rate coefficient is
similar to the value found for $d_{34}$OA (Table 2), $\tau_{d,NO3,1}$ is double of the value found for oleic acid, this lifetime
refers to the monolayer when is highly packed (see description in Section 2.2) and that is the condition where the
difference in chain length between $d_{14}$POA and $d_{34}$OA can play a role. The higher value of $\tau_{d,NO3,1}$ for $d_{14}$POA is
consistent with the hypothesis of an easier access to the double bond due to the shorter alkyl chain of $d_{14}$POA.
The $\tau_{d,NO3,2}$ does not show a big difference between $d_{14}$POA and $d_{34}$OA and that refers to the monolayer in a less
dense state, suggesting that once the access to the double bond is comparable the reaction has a similar
behaviour for the two molecules. $d_{14}$POA surface excess data have larger experimental errors than the fully
deuterated molecules.
**3.3. Methyl oleate ($d_{33}$MO) exposed to nitrate radicals (NO$_3$)**
Methyl oleate possesses the same aliphatic chain as OA, but it has a different head group: instead of a carboxylic
acid it has a methyl ester (COOCH$_3$) group. Fully deuterated $d_{33}$MO was used (see Table 1 in the ESI). MO
occupies a larger surface area and is less stable at the air–water interface than OA because of its less hydrophilic
head group (see isotherm in Section 1 of the ESI). However, the reactive site is in a similar chemical
environment as for OA, and any difference in reaction kinetics is expected to be related to the chain orientation
and formation of different products.
Figure 8 displays the surface excess decays of $d_{33}$MO monolayers at the air–ACMW interface as a function of
time with respect to [NO$_3$]. [NO$_3$] was varied from $(13 \pm 6)$ ppt to $(86 \pm 45)$ ppt.

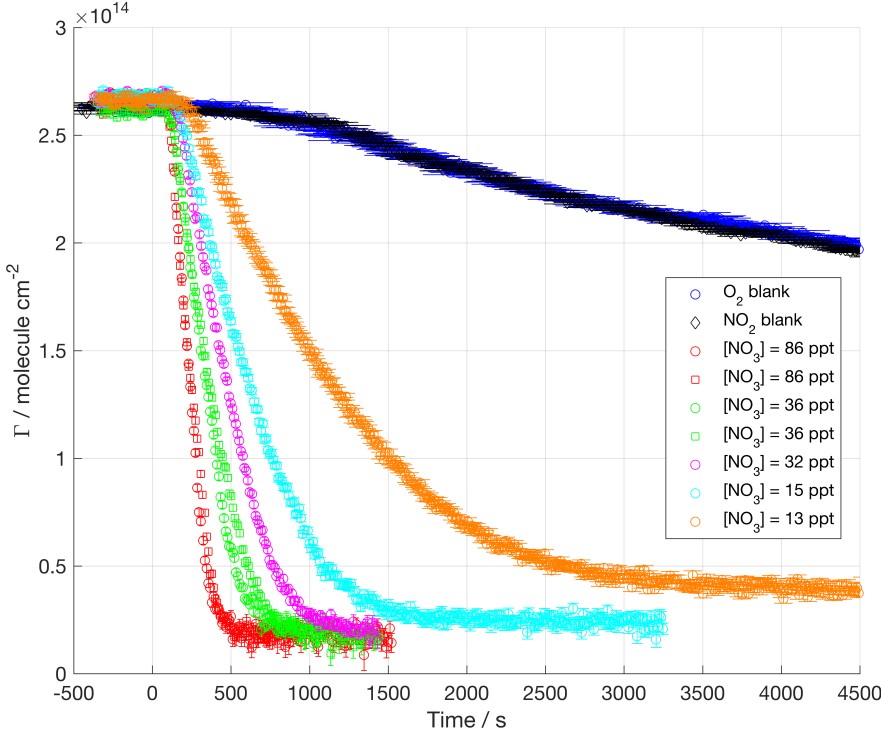

**Figure 8.** Surface excess of methyl oleate ($d_{33}$MO) exposed to different [NO$_3$], mean values are displayed in the legend. NO$_3$ exposure is started at $t = 0$ s.

The kinetic decays presented in Figure 8 show a very clear dependence on [NO$_3$] and very good signal-to-noise ratios. The decays are generally faster than for both $d_{34}$OA and $d_{14}$POA. The exposure to O$_2$ and NO$_2$ flow leads to similar surface excess decays; this non-reactive loss is significantly larger than those recorded for $d_{34}$OA and $d_{14}$POA suggesting that $d_{33}$MO is not as stable at the air–water interface as $d_{34}$OA and $d_{14}$POA. The apparent rate coefficient obtained for the decays in absence of NO$_3$ is about $2 \times 10^{-10}$ molecule cm$^{-2}$ s$^{-1}$. As for $d_{34}$OA, the reaction starts with a slightly increasing delay as the oxidant concentration is lower; the formation of droplets floating on top of the monolayer after spreading could explain this effect, since the compound is liquid at room temperature and evidence of lenses was found in BAM images (see Section 1 in the ESI). The minimum value reached by the surface excess is $\approx 2 \times 10^{13}$ molecule cm$^{-2}$, which is at the detection limit. Therefore, no surface-active products are expected to remain at the interface as was also found in ozonolysis experiments with $d_{33}$MO in the same chamber (Sebastiani et al., 2015); this was also confirmed by complementary ellipsometry measurements in the same reaction chamber (data not shown). According to this finding, the product yields were chosen as follows: $c_S = 0.03$, $c_G = 0.45$ and $c_B = 0.52$. The $c_s$ value was set to 0.03 in order to account for the surface excess detection limit considering the experimental background. The kinetic parameters were constrained to the following value ranges: $k_{surf}$ was allowed to vary $(0.7 - 4) \times 10^{-8}$ cm$^2$ molecule$^{-1}$ s$^{-1}$, $\tau_{d,NO3,1}$ varied between $(5 - 20) \times 10^{-9}$ s, $\tau_{d,NO3,2}$ varied between $(10 - 50) \times 10^{-9}$ s and $\tau_{d,NO2}$ varied between $(0.1 - 6) \times 10^{-8}$ s (see Table 2). An example of the fitting resulting from the kinetic modelling is displayed in Figure 9. The best-fit values obtained from the kinetic model are presented in Table 2. The rate coefficient for $d_{33}$MO is slightly larger than those for both $d_{34}$OA and $d_{14}$POA, while the desorption times are similar to those found for $d_{34}$OA and $d_{14}$POA with the exception of the doubled $\tau_{d,NO3,1}$ for POA further confirming the better accessibility of the double bond for the shorter chained POA compared to both OA and MO. All fits are presented in the ESI.

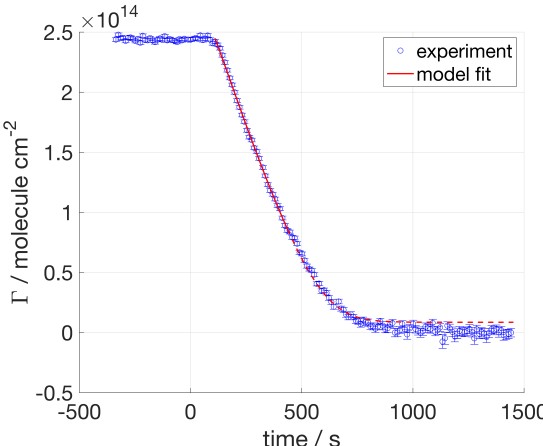

**Figure 9.** Methyl oleate ($d_{33}$MO) exposed to [NO$_3$] = 36 ppt.  The red line illustrates the fit obtained from our kinetic modelling (the solid section of the line indicates the data range used for the kinetic analysis; the dashed section of the model line illustrates the calculated final part of the decay, but the corresponding experimental data were not used in the optimisation of the fitting).

### 3.4 Stearic acid ($d_{35}$SA) exposed to nitrate radicals (NO$_3$)

In addition to adding to the double bond of the unsaturated surfactants discussed in the previous sections, NO$_3$ may abstract hydrogen atoms from the aliphatic tail (Shastri & Huie, 1990; Wayne et al., 1991; Mora-Diez et al., 2002). In order to investigate the contribution of this hydrogen abstraction, the saturated surfactant stearic acid was exposed to NO$_3$. Figure 10 shows the comparison between the surface excess of a $d_{35}$SA monolayer exposed to O$_2$ and to NO$_3$ at (86 ± 45) ppt.

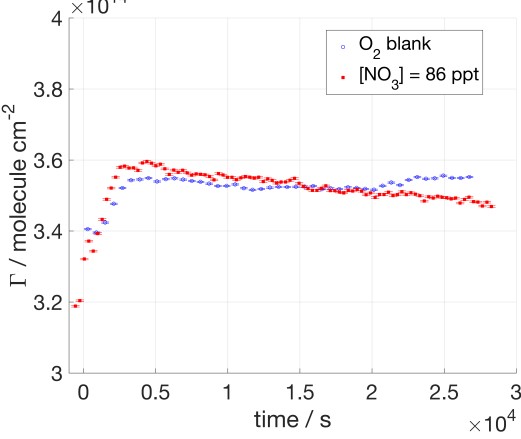

**Figure 10.** Surface excess of stearic acid ($d_{35}$SA) exposed to O$_2$ (blue circles) and to [NO$_3$] = 86 ppt (red filled squares). Exposure to NO$_3$ starts at $t$ = 0 s. Both surface excess traces show an increase over the first 40 min. There is slight subsequent decrease in the surface excess during exposure to NO$_3$.

The data were recorded for more than 8 h for both gas-phase environments. The initial surface excess evolution of the monolayer exposed to NO$_3$ is comparable to that for the O$_2$ blank: both profiles show a slow increase in surface excess in the first 40 min. Apart from the initial increase in $\Gamma(t)$ values, no measurable change in the surface excess has been recorded when SA is exposed to O$_2$, and the film is shown to be stable on the probed time scale; in presence of NO$_3$ a slight decrease in surface excess hints at a slow reactive decay. From these data we estimated a rate coefficient, $k_{surf}$, of approximately (5 ± 1) × 10$^{-12}$ cm$^2$ molecule$^{-1}$ s$^{-1}$; the parameters ranges and initial values in the model were kept as for OA for consistency, because of the lack of any experimental data

on products and very limited kinetic data due to the very slow process; the lower limit for the rate coefficient
was decreased to $1 \times 10^{-12}$ cm$^2$ molecule$^{-1}$ s$^{-1}$; the model fit to the experimental data is displayed in the ESI; the
estimated kinetic parameters should be considered with caution given the severe limitations mainly due to the
lack of experimental data. For this system, the surface coverage never reached below 90% of the initial value and
hence the determination of the second desorption times, $\tau_{d,NO3,2}$, is not accurate (the value obtained for $\tau_{d,NO3,2}$
actually corresponds to the lower limit of the constrained range; see value in square brackets in Table 2). It
should be noted that in our experimental approach it is theoretically possible that the chemical composition of
the monolayer could change upon reaction with NO$_3$ (e.g. formation of organonitrates; Gross & Bertram, 2009)
while the scattering excess (i.e. the product of $\rho$ and $d$ in Eq. (2)) could by coincidence remain unchanged during
this process; the resulting $\Gamma(t)$ plot would then also remain constant. This is highly unlikely, in particular since
our result is in accordance with the findings of Knopf et al. (2006), where the exposure to [NO$_3$] = 100 ppt for
one week resulted in a maximum of 10% of the organic monolayer being volatilised (the monolayer was
supported on a solid substrate and the measurement does not rely on the neutron scattering length density).  For
practical reasons it is not feasible to carry out NR experiments on a similar time scale, however our results
suggest that the kinetic behaviour may be affected by the type of substrate given the faster oxidation of $d_{35}$SA
observed at the air–water interface during exposure to NO$_3$.
**4. Discussion**
The kinetic parameters obtained by analysing the NR data allow investigation of the effects of the chemical
structure, i.e. chain length, degree of unsaturation and head group properties. A summary of the kinetic results
reported in the present study is given in Table 3. For the unsaturated molecules studied we obtained rate
coefficients in the order of $10^{-8}$ cm$^2$ molecule$^{-1}$ s$^{-1}$, which leads to uptake coefficients, $\gamma$, for NO$_3$ on a droplet
covered in a monolayer of organic compound to be in the order of $10^{-3}$. These results broadly agree with the very
limited number of measurements found in the literature (Moise et al., 2002; Knopf et al., 2006; Gross &
Bertram, 2009; Xiao & Bertram, 2011; Zhao et al, 2011; Zhang et al., 2014b) for unsaturated organics exposed
to NO$_3$ in particular when considering that experiments are often carried out in very different conditions (e.g. on
a gold surface instead of the water surface we used) and employ fundamentally different experimental
approaches (e.g. flow tubes).  Moise et al. (2002) studied the uptake of NO$_3$ by a range of liquid or frozen
organics in a rotating wall flow tube, and they measured uptake between $1.6 \times 10^{-3}$ and $1.5 \times 10^{-2}$ depending on
the kind of liquid organic compounds. Gross & Bertram (2009) determined the uptake of NO$_3$ by a self-
assembled alkene monolayer at the solid substrate obtaining an uptake coefficient of 0.034. They suggested that
a possible reason for this higher value compared to the results of Moise et al. (2002) is the location of the double
bond at the interface. Zhang et al. (2014b) determined the uptake coefficient of NO$_3$ on a model surface of a self-
assembled monolayer of vinyl-terminated alkanethiols on gold substrate to be $(2.3 \pm 0.5) \times 10^{-3}$ monitoring the
double bond rupture. The present results for organic monolayers at the air–water interface are in a better
agreement with those of Moise et al. (2002) and Zhang et al. (2014b). The agreement with Moise et al. (2002)
may suggest that the accessibility of the reactive site for these monolayers is similar to that of a thick film.
However, the work of Zhang et al. (2014b) was on an organic monolayer at the air–solid interface and the rate of
product formation was measured instead of the NO$_3$ consumption as in Gross & Bertram (2009); in a way our
approach is closer to that of Zhang et al. (2014b), since we followed the organic reactant loss *in situ*. Given the

complex chemical environments these surfactants will encounter in the atmosphere it would be important to investigate the difference in uptake coefficients of $NO_3$ by organic monolayers adsorbed to different substrates and compare uptake coefficients based on both consumption of $NO_3$ and product formation rates. King et al. (2009) investigated OA oxidation by $O_3$ on different subphases with pH ranging from 2 to 7 and no significant change was found in the rate coefficient. In our experiments with the oxidant $NO_3$ we expect $HNO_3$ to be formed, and induce a change in pH in the subphase, but given the fact that it was previously reported that there was no pH effect, we did not explore the pH changes in the present study.

**Table 3.** Kinetic parameters, uptake coefficients and estimated monolayer lifetimes for the compounds studied. Literature values for uptake coefficients on similar compounds are included for comparison.

| Surfactant | $k_{surf}$ / cm$^2$ molecule$^{-1}$ s$^{-1}$ | $\gamma$ / 10$^3$ | $\gamma_{lit}$ / 10$^3$ | Lifetime[a] |
|---|---|---|---|---|
| $d_{35}$SA | $(5 \pm 1) \times 10^{-12}$ | $(5 \pm 1) \times 10^{-4}$ | $(8.8 \pm 2.5) \times 10^{-1}$ [b] | 21 days |
| $d_{34}$OA | $(2.8 \pm 0.7) \times 10^{-8}$ | $2.1 \pm 0.5$ | $(3 \pm 1) \times 10^2$ [c] <br> $[1.6 \pm 0.3]$ [d] | 6 minutes |
| $d_{14}$POA | $(2.4 \pm 0.5) \times 10^{-8}$ | $1.7 \pm 0.3$ | $[2.3 \pm 0.5]$ [e] <br> $[34^{+44}_{-18}]$ [f] | 7 minutes |
| $d_{33}$MO | $(3.3 \pm 0.6) \times 10^{-8}$ | $2.1 \pm 0.4$ | $[(1.4^{+8.6}_{-0.5}) \times 10^2]$ [g] | 5 minutes |

[a] see Section 4.3 for details on the lifetime calculation;
[b] value refers to a self-assembled monolayer on a gold substrate (Knopf et al., 2006);
[c] value refers to a study with a flow tube coupled to a chemical ionisation mass spectrometer (Zhao et al., 2011);
[d] value refers to 1-octadecene uptake measured in a rotating wall flow tube (Moise et al., 2002);
[e] value refers to a vinyl-terminated self-assembled monolayer at a gold surface, which was chosen as a model for a double bond positioned at the gas–surface interface by Zhang et al. (2014b);
[f] value refers to a terminal alkene monolayer at a gold surface (Gross & Bertram, 2009);
[g] value refers to binary mixtures of MO and saturated molecules measured in a rotating wall flow tube (Xiao & Bertram, 2011).

The products yields used in our model were based on the findings of Docherty & Ziemann (2006) and Hung et al. (2005); both papers present possible mechanisms for product formation from the oleic acid droplets reacting with $NO_3$ in presence of $O_2$ and $NO_2$. $NO_3$ attacks the double bond and the primary reaction is most likely to lead to the formation of an organonitrate, which would maintain the $C_{18}$ chain instead of splitting into $C_9$ fragments; however, subsequent reactions have been found to lead to shorter molecules, such as nonanal and 9-oxononanoic acid (Docherty & Ziemann, 2006). Organonitrates are reactive species that are likely to undergo further reactions and produce smaller fragments, which either are lost to the gas- or water-phase or remain at the interface. In previous work (Hung et al., 2005; Docherty & Ziemann, 2006), the primary organonitrates were found to be more abundant than shorter fragments, but these studies focused mostly on the first few seconds to minutes of the reactive degradation, while our work on unsaturated surfactants follows the reaction until the organic film is fully processed. The surface-active products were found to total 20% and 15% (based on the deuterated proportion of the molecule only) of the initial amounts of $d_{34}$OA and $d_{14}$POA, while $d_{33}$MO does not lead to any surface-active products ($\leq$ 3%), probably due to the lower surface activity of the COOCH$_3$ head group. The proportion of volatile and soluble products is mainly based on solubility and volatility estimations (Kuhne et al., 1995; Compernolle et al., 2011); this distinction was used to predict the time evolution of the concentrations of these products and their contribution to the surface excess when produced at the interface. $d_{14}$POA is expected to behave similarly to $d_{34}$OA, except the formation of $C_8$ fragments with slightly higher solubility & volatility and hence a decreased surface-active yield; to our knowledge no studies on $d_{14}$POA exposed to $NO_3$ were performed

and no data are available on the products formed. For all the reactions studied here we expect secondary reactions not to be significant due to our set-up with a one-molecule thin layer of organic molecules each containing only a single reactive site ($NO_3$-initiated hydrogen abstraction is much slower than addition to the double bond as demonstrated in our work on the oxidation of the saturated surfactant stearic acid). Multiple generations of oxidation products could not be resolved in this experimental approach and are not considered explicitly in this work. Simultaneous neutron reflectometry and infrared reflection absorption spectroscopy (IRRAS), a technique we have recently developed for study of related systems (Skoda et al., 2017) may be able to give some information on the chemical composition of one-molecule thin films during kinetic studies of oxidation reactions at the air–water interface in the future.

Although our present approach did not allow convenient variation of the surface excess due to the barrier-less Langmuir trough in our miniature kinetic chamber optimised for kinetic measurements of fast reactions (Sebastiani et al., 2015), we believe that the best fit parameters we report in the present study can predict the fate of an organic monolayer with a different compression, i.e. at a different initial surface excess.

The key findings of the present work in relation to surfactant chain length, head group and saturation are discussed in the following paragraphs.

**4.1. Chain length**

The slightly lower rate coefficient of $d_{14}$POA compared to $d_{34}$OA is hard to rationalise (the rate coefficients obtained overlap with the experimental uncertainties), since –if anything– we would have expected $d_{14}$POA to react slightly faster given the fact that the two molecules are identical except a shorter alkyl chain that could facilitate attack of $NO_3$ in the case of $d_{14}$POA (as seems to be the case for $O_3$ attack on OA and POA in a complex 12-component mixture containing these two compounds: Huff Hartz et al. (2007) reported ratios of effective condensed phase rate constants of $7 \pm 3$ and $6 \pm 2$ for POA and OA ozonolysis, respectively; no kinetic measurements have been reported for the $d_{14}$POA–$O_3$ system to our knowledge). However, the reactivity depends on the desorption time as well (Table 2); the longer the lifetime of adsorption, the higher is the possibility to react; $\tau_{d,NO3,1}$ for $d_{14}$POA is double the value found for $d_{34}$OA, which confirms the hypothesis of an easier access to the double bond due to the shorter alkyl chain of $d_{14}$POA.

The uncertainty of the rate coefficient corresponds to the standard deviation of the values found for the rate coefficients for each oxidant concentration; a lower uncertainty means that the values obtained from the different oxidant concentrations are closer to each other. Since the rate coefficients obtained for the individual experiments for $d_{14}$POA agree slightly better than those for the other surfactant reactions, a smaller $\chi^2_\nu$ is obtained despite the clearly visible scatter in the $d_{14}$POA surface excess profiles (see Fig. 6) and the larger error bars on the data.

**4.2. Head group**

The rate coefficients displayed in the second column of Table 3 for the reactions with $NO_3$ show a small, but statistically significant difference between the unsaturated organic compounds investigated: $d_{33}$MO reacts

slightly faster than $d_{34}$OA with $d_{14}$POA reacting the slowest. This order of reactivity is broadly consistent with
that found for the ozonolysis of $d$MO (Pfrang et al., 2014; Sebastiani, et al., 2015) and $d_{34}$OA (King et al., 2009)
at the air–water interface, but the differences are less pronounced for the more reactive NO$_3$: $k_{surf,NO3}$ / $k_{surf,O3}$
ratios are ~ 384 and ~ 58 for $d_{34}$OA and $d_{33}$MO, respectively.
A direct comparison between surface excess decays for the three unsaturated surfactants allows us also to
examine if there is a correlation between the type of head group and the presence of products at the air–water
interface. Molecules with a fatty acid (COOH) head group (i.e. $d_{34}$OA and $d_{14}$POA) left a considerable
proportion of surface-active products at the air–water interface, while $d_{33}$MO with its methyl ester (COOCH$_3$)
head group did not leave any detectable product ($\leq$ 3% surface-active products based on the detection limit for
our experimental set-up). Therefore, the retention of the organic character at the air–water interface differs
fundamentally between the different surfactant species: the fatty acids studied form products with a yield of ~
20% that are stable at the air–water interface while the NO$_3$-initiated oxidation of the methyl ester rapidly
removes the organic character from the surface of the aqueous droplet. A similar difference (King et al., 2009;
Pfrang et al., 2014; Sebastiani et al., 2015) between methyl ester and parent fatty acid has been found for the
ozonolysis of $d_{34}$OA and $d_{33}$MO, but the retention of 20% of organic material at the air–water interface is even
more surprising for the more highly reactive nitrate radicals. The film-forming potential of the reaction products
thus strongly depends on the head group properties.
**4.3. Chain saturation**
Unsurprisingly, the fate of the monolayer is altered fundamentally by the absence of unsaturation in the aliphatic
chain. In fact, $d_{35}$SA loss from the interface during our 8 h experiments was extremely small, while the initial 40
minutes of reaction lead to an increase of surface excess for both NO$_3$ and O$_2$. An increase in surface excess may
depend on a closer packing of the aliphatic chains that is more likely than gas-phase species absorbing to the
interface, since gas absorption was not found for the other molecules studied. Indeed, we have recently reported
an apparent increase in NR signal most likely caused by changes in the structure at the air–water interface for a
two-component mixture of immiscible surfactants (Skoda et al., 2017). Our implementation of NR only at low-$q$
provides a measure of the total neutron scattering excess rather than a direct measure of the surface excess of the
organic material at the interface hence there is a possibility that the film composition may be changing over time
due to gas adsorption into the monolayer, e.g. formation of organonitrates by NO$_3$ (Gross & Bertram, 2009). Due
to limited access to neutron beam time, only one experiment was performed on $d_{35}$SA lasting 8 h and it led to an
estimation of the rate coefficient of $(5 \pm 1) \times 10^{-12}$ cm$^2$ molecule$^{-1}$ s$^{-1}$, which is four orders of magnitude lower
than the rate coefficient for the unsaturated molecules. This value has to be considered with caution, since it
relies on the modelling of only one data set, corresponding to the highest NO$_3$ concentration, and the parameters
in the modelling were the same as for $d_{34}$OA except for the lower limit of the rate coefficient that has been
reduced to $1 \times 10^{-12}$ cm$^2$ molecule$^{-1}$ s$^{-1}$. This was necessary because of the lack of previous experimental data to
constrain the model and the limited reaction extent that could be observed during the available beam time.

The higher stability of SA monolayers upon oxidation compared to the unsaturated molecules suggests that SA
may concentrate at the aerosol surface leading to a stabilisation of the particles. Formation of such a stable film

may protect more reactive species, located within the aerosol bulk (Pfrang et al., 2011), by slowing down the diffusion of the organic compound from bulk to surface and the diffusion of the oxidant from the gas phase to the bulk. Accumulation of saturated films in aged organic films has indeed recently been reported (Jones et al., 2017).

### 4.4. Atmospheric implications

Contrasting the oxidation of $d_{33}$MO upon exposure to $O_3$ (Pfrang et al., 2014; Sebastiani, et al., 2015) and $NO_3$ shows –as expected– a clearly stronger oxidative power of $NO_3$ compared to $O_3$. The oxidative power may be quantified from the uptake coefficient (Gross & Bertram, 2009) of $NO_3$ and $O_3$ as the product of uptake coefficient and gas-phase oxidant concentration. $O_3$ is found in the atmosphere at concentration between 10 and 100 ppb. The oxidative power calculated for the lowest concentration would be $7.5 \times 10^6$ molecule $cm^{-3}$. For the calculation of the oxidative power, $[NO_3]$ was chosen to be representative of a range of atmospheric mixing ratios (5–50 ppt, i.e. ca. $1.4–13.5 \times 10^8$ molecule $cm^{-3}$), which could be encountered in the atmosphere owing to spatial and seasonal fluctuations (Seinfeld & Pandis, 2006). The resulting oxidative powers are $1.2 \times 10^6$ molecule $cm^{-3}$ and $12 \times 10^6$ molecule $cm^{-3}$ for lowest and highest $[NO_3]$, respectively. Although the concentration of $NO_3$ in the atmosphere is low compared to $[O_3]$, our results suggest that night-time oxidation is likely to be often dominated by $NO_3$-initiated degradation. This finding suggests that further investigation of the oxidation driven by $NO_3$ is required to understand the fate of aerosol droplets together with studies of the key daytime oxidant OH. This conclusion is also supported by a very recent study (Jones et al., 2017) suggesting that atmospheric surfactants are essential inert with respect to ozonolysis making studies of $NO_3$ as well as OH-initiated oxidation even more timely.

The lifetime of an organic monolayer is calculated (Moise & Rudich, 2001; Knopf et al., 2011) as the inverse of the product of $k_{surf}$ and $[NO_3]_s$, the $NO_3$ surface concentration was calculated as in Smith et al. (2002) using a $[NO_3] = 20$ ppt ($5.4 \times 10^8$ molecule $cm^{-3}$). Based on our kinetic experiments, the lifetime with respect to $NO_3$-initiated oxidation of an organic monolayer of monounsaturated molecules with a surface concentration of $3 \times 10^{14}$ molecule $cm^{-2}$ on an aqueous droplet is ca. 5 to 7 minutes, while it becomes about 21 days for saturated species. Zhao et al. (2011) estimated for a 100 nm droplet of pure oleic acid exposed to 25 ppt $NO_3$ a lifetime of ca. 35 minutes. The direct comparison with our kinetic study on a self-assembled monolayer at the air–water interface suggests that oleic acid molecules in a pure oleic acid droplet would be degraded ca. 20 times faster than the same number of oleic acid molecules present in a self-assembled monolayer at the air–water interface of an aqueous droplet. Self-assembly thus may play a significant role for the kinetic behaviour of surfactant molecules in the atmosphere. We are currently carrying out experimental studies on oleic-acid based aerosol proxies with complementary techniques (Seddon et al., 2016) to further investigate the importance of complex self-assembly in atmospheric aerosols (Pfrang et al., 2017).

The loss of the organic character from the air–water interface will have consequences for the surface tension of aqueous droplets in the atmosphere: an organic surfactant film substantially reduces the droplet's surface tension compared to pure water, so that the film-forming potential of degradation products of these surfactant films is of key interest. We found that the stability of products formed at the air–water interface differs substantially

between the fatty acids (OA and POA) and the methyl ester (MO) studied. The head group thus seems key to
determine whether the surfactant will be able to reduce the surface tension of water droplets for any considerable
time which could have important consequences for droplet growth and should be considered when developing
emission control strategies.
The rapid loss of the organic monolayers at the air–water interface demonstrated by our experimental data of the
oxidative decays is surprising given a number of field studies reporting much longer residence times of
unsaturated surfactants in atmospheric aerosols (Morris et al., 2002; Knopf et al., 2005; Ziemann, 2005; Zahardis
& Petrucci, 2007). Such unsaturated organics may have longer lifetimes if protected from oxidative attack by
gas-phase species e.g. inside highly viscous aerosol particles (Virtanen et al., 2010; Pfrang et al., 2011; Shiraiwa
et al., 2011; Shiraiwa et al., 2013) or if mixed with non-reactive species in a complex surface film with yet
unexplored kinetic behaviour. This provides a key motivation to investigate the oxidation of mixed surfactant
films, which represent closer proxies for real atmospheric aerosol droplets in the future. These measurements
have commenced already in our group, and as such the findings presented here provide an essential experimental
basis for an extension of the work and methodology towards an improved understanding of the complex
behaviour of atmospheric aerosols.
**5. Conclusions**
We have investigated the reactions of the key atmospheric oxidant $NO_3$ with organic monolayers at the air–water
interface as proxies for the night-time ageing of organic-coated aqueous aerosols. The surfactant molecules
chosen allowed the investigation of the effects of chain length, head group properties and degree of unsaturation
on the reaction kinetics as well as the proportion of surface-active products formed. The experimental results
presented together with the tailored modelling approach for the four structurally different monolayers has
allowed determination of the kinetic parameters of heterogeneous reactions at the air–water interface with $NO_3$
for the first time. The study of heterogeneous reactions of organic monolayers at the air–water interface exposed
to oxidants is crucial to understand the role of such films for the atmospheric fate of organic-coated aqueous
aerosols (Gilman et al., 2004). Previous studies performed on these types of reactions were nearly exclusively
carried out monitoring the gas-phase species (Wadia et al., 2000; Knopf et al., 2007; Cosman et al., 2008a;
Cosman et al., 2008b). Gross & Bertram (2009) investigated the oxidation of organic monolayers at an air–solid
interface and in addition to monitoring the gas-phase species during the reaction, they analysed the product film
with several surface spectroscopic techniques. The monitoring of the organic monolayer during oxidation at the
air–water interface was introduced by King et al. (2009) for the study of OA exposed to $O_3$. To the best of our
knowledge, no-one has previously investigated the oxidation of organic monolayer at the air–water interface by
$NO_3$ by in situ kinetic measurements of the surface excess.

NR experiments together with tailored kinetic modelling allowed us to determine the rate coefficients for the
oxidation of OA, POA and MO monolayers to be $(2.8 \pm 0.7) \times 10^{-8}$ cm$^2$ molecule$^{-1}$ s$^{-1}$, $(2.4 \pm 0.5) \times 10^{-8}$ cm$^2$
molecule$^{-1}$ s$^{-1}$ and $(3.3 \pm 0.6) \times 10^{-8}$ cm$^2$ molecule$^{-1}$ s$^{-1}$, for fitted initial desorption lifetimes of $NO_3$ at the
closely packed organic monolayers, $\tau_{d,NO3,1}$, of $8.1 \pm 4.0$ ns, $16 \pm 4.0$ ns and $8.1 \pm 3.0$ ns, respectively. The
approximately doubled desorption lifetime found in the best fit for POA compared to OA & MO is consistent

with a more accessible double bond associated with the shorter alkyl chain of POA facilitating initial $NO_3$ attack at the double bond in a closely packed monolayer. The corresponding uptake coefficients for OA, POA and MO were found to be $(2.1 \pm 0.5) \times 10^{-3}$, $(1.7 \pm 0.3) \times 10^{-3}$ and $(2.1 \pm 0.4) \times 10^{-3}$. For the much slower $NO_3$-initiated oxidation of the saturated surfactant SA we estimated a rate coefficient of approximately $(5 \pm 1) \times 10^{-12}$ $cm^2$ molecule$^{-1}$ s$^{-1}$ leading to an uptake coefficient of approximately $(5 \pm 1) \times 10^{-7}$.

Our investigations demonstrate that $NO_3$ will make a substantial contribution to the processing of unsaturated surfactants at the air–water interface during the night given its reactivity is ca. two orders of magnitude higher than that of $O_3$. Furthermore, the relative contributions of $NO_3$ and $O_3$ to the oxidative losses vary massively between structurally closely related species: $NO_3$ reacts ~ 384 times faster than $O_3$ with the most common model surfactant OA, but only ~ 58 times faster with its methyl ester MO. It is therefore required to perform a case-by-case assessment of the relative contributions of the different degradation routes for any specific surfactant. The impact of $NO_3$ on the fate of saturated surfactants is slightly less well quantified given the limited kinetic data, but $NO_3$ is very likely to be a key contributor to the loss of saturated species at night-time taking over from OH-dominated loss during the day.

The retention of the organic character at the air–water interface also differs fundamentally between the surfactant species studied. On the one hand, the fatty acids (OA and POA) form products stable at the air–water interface with yields of ~ 15–20%. On the other hand, $NO_3$-initiated oxidation of the oleic acid methyl ester MO rapidly removes the organic character from the surface of the aqueous droplet ($\leq 3\%$ surface-active products). The film-forming potential of reaction products will thus depend on the relative proportions of saturated and unsaturated surfactants as well as the head group properties.

The lifetime with respect to $NO_3$-initiated oxidation of an organic monolayer of monounsaturated molecules is about 5 to 7 minutes, while it becomes about 21 days for saturated species. Actual atmospheric residence times of unsaturated species are much longer than the lifetimes determined with respect to their reactions at the air–water interface, so it follows that they must be protected from oxidative attack *e.g.* by incorporation into a complex aerosol matrix or in mixed surface films with yet unexplored kinetic behaviour.

**Acknowledgements**

The authors are grateful to Prof. Ulrich Pöschl and Dr Manabu Shiraiwa for expert advice on the PRA modelling. The authors would like to thank Dr Francesco Piscitelli and Dr Ernesto Scoppola for the help during the night shifts on FIGARO. We would like to thank the Partnership for Soft Condensed Matter for access to the ellipsometer, and the ILL (Grenoble, France) for allocations of beam time on FIGARO. FS is grateful for support from the ILL and the University of Reading in the framework of the NEATNOx studentship. KR is grateful to NERC for his studentship. CP thanks NERC (grant number NE/G000883/1) for support.

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
