# Peer review of "Night-time oxidation of surfactants at the air-water interface: effects of chain length, head group and saturation."

_Atmospheric Chemistry and Physics, 2017_

## Referee Comment (RC1) · Anonymous Referee #1 · 4 Sep 2017

In this work the authors explore the oxidation of unsaturated surface active organic molecules (and one saturated organic molecule) by NO3 radicals. They report data using neutron reflectometry and show the decay of signal with increasing reaction time. Kinetic models were applied to infer oxidant concentrations and to interpret the observations in light of a series of surface and near-surface processes. The investigation is interesting and has some relevance to oxidative processing of films on the ocean surface, aerosol particles and cloud droplets. In the current version of the manuscript, some severe limitations exist that call into question the usefulness of the kinetic parameters. I hope that in addressing the points below the authors can provide a more compelling description of their work and the validity of their conclusions.

[Figure]

Main points:

1) The methodology associated with preparing the film should be detailed in the main text.

2) As far as I can tell, the compression of the film for the oxidation experiments is not reported anywhere. Please amend or make this information more prominent in the main text.

3) A more basic introduction to the NR technique is necessary, focusing on the observed quantities and what this actually means in these experiments. Where does the value of 'd' come from? Does 'd' change during an experiment in which shorter chain surface active species may be created, and how is this accounted for?

4) Uptake of N2O5 into aqueous phases can lead to acidification as HNO3 is formed. How does the pH of the aqueous sub-phase change during the measurements, and how might changes in pH affect the film properties? Could changes in pH contribute towards the plateau observed in the initial time during some measurements? Might there be competition between uptake of N2O5 and NO3?

5) It is not clear what assumptions are made in order to derive the rate constants. In particular, the authors should perform a sensitivity analysis to see how changes in branching ratios affect the results.

6) Looking at the data for OA vs POA, the uptake coefficients are similar, but the time constants are a factor of 2 different. What causes this? There appears to be no relation to NO2/NO3 adsorption lifetimes and uptake, so what purpose do they serve in the model? How would a change in adsorption lifetime manifest itself in the experimental data or the parameters they pull out?

7) What would the decay curves look like if all the products remained at the surface? Given that the technique can only provide information on the partitioning of products away from the surface, how can the authors be sure that multiple generations of oxidation are not occurring prior to material desorbing from the interface?

8) The modelling is performed in such as way that it is not clear if there is any predictive power to the results. For example, the model is fit to the initial decay, and then floated for the remaining time, and in most cases this free-floating region does not do a good job as describing the data. Is this because additional processes are occurring that are not factored into the model? Can the parameters obtained be used to accurately predict the chemistry at different film compressions, oxidant concentration etc.?

9) In the model, partitioning away from the surface (either into the gas phase or bulk aqueous solution) is rapid. Is there any consequence in the modelling for partitioning to one or the other? In these experiments, would the same results be obtained if the products were simply broken down into surface-present and surface-absent? How are partially surface active molecules accounted for in the present analysis?

10) The data for stearic acid is not convincing – while it clearly shows less reactivity, the magnitude of the decay is very small. I would suggest this be removed or moved to the SI to allow for the additional material in the main text to address the previous points in this review.

Minor points:

1) A figure in the main text showing the structure of the molecules would make comparisons of the datasets easier for a reader.

2) The reference to Section 4 of the SI containing examples of raw data is wrong – there is no raw data presented in the SI.

3) Consistency of units (some mixing of [m] and [cm] between text and figures when reporting surface excess.

4) Define acronyms consistently, even if they are well known in your field (e.g. FIGARO, PRA)

5) Define the term "surface excess" – it is unclear in the derivation if this is actually a surface concentration, as equation 2 seems to indicate. If I am correct in my understanding of the difference, for very insoluble species surface concentration and surface excess are approximately equal, but please clarify this.

6) The first paragraph of the "Discussion" would be more appropriate in either the Introduction or the Conclusions.

7) Slightly excessive 'keywording'

---

## Referee Comment (RC2) · Anonymous Referee #2 · 11 Sep 2017

Major The authors give the relevance of the manuscript towards insight into organic coated aerosol (first sentence in the abstract). However, the experimental data is not taken from aerosol but from a flat surface, which could have a link to the marine boundary layer. This is significant given the known deviation of aerosols from the behaviour expected from a flat surface, that is more pronounced as the droplet size decreases and surface curvature increases. I would expect there to be a significant effect due to change in surface tension that would alter the kinetics between aerosol and flat surfaces that makes a flat surface an unsuitable experimental model for aerosol. Although the authors do note on page 21 that they are currently looking at aerosol proxies, none of this data appears to be included in the manuscript. The experimental system has

been previously outlined in Sebastiani, RSC Adv (2015), and is highly relevant to this manuscript. But even so, there does need to be more in-depth discussion of the experimental techniques used here particularly where these form the basis for drawing conclusion or inference. As the manuscript stands very little detail is given on the experimental setup and measurement, and crucially the conditions that measurements are taken under. Has the effect of NO3 flow rate on oxidation been considered? From page 4 there seems to be a range used, but details are not given with each measurement. On page 5 the authors mention the laser alignment window, which seems to be for a 632.8 nm Helium Neon laser mentioned in Sebastiani, RSC Adv (2015). Is this experimental setup appropriate for this measurement given the intention to measure night-time oxidation? There needs to be further discussion to establish the conditions that the samples were kept under, e.g. was a dark room used to prepare samples, or an indication is needed that this has been considered. Further discussion on the potential products that could form should be included. Without compositional information the assumptions made during the oxidation process and kinetic analysis are not convincing. Have the authors assumed that the products will not undergo further oxidation or degradation?

Minor Figure 4 appears before table 1, but look to be discussed in a different order in the text. Acronyms such as BAM should be written in full. Symbols used in equations should be explained for clarity in the text, e.g. in section 2.1.3; lambda and theta. Page 13: The variable parameters on line 13 should have "is", an equals sign, "varied between", or something appropriate between symbol and values.

---

## Author Comment (AC1) · 6 Nov 2017

Please find attached our detailed response to all reviewers' comments together with a revised version of the manuscript with the changes and additions tracked.

We have attached rather than inserted our response since the complex formatting of our 18-page response would get lost thus making the document much harder to read.

Please also note the supplement to this comment:
https://www.atmos-chem-phys-discuss.net/acp-2017-651/acp-2017-651-AC1-supplement.zip

---

## Author Response (AR1)

**Referee #1**

*"In this work the authors explore the oxidation of unsaturated surface active organic molecules (and one saturated organic molecule) by NO3 radicals. They report data using neutron reflectometry and show the decay of signal with increasing reaction time. Kinetic models were applied to infer oxidant concentrations and to interpret the observations in light of a series of surface and near-surface processes. The investigation is interesting and has some relevance to oxidative processing of films on the ocean surface, aerosol particles and cloud droplets."*

We are grateful for the referee's positive comments on our work.

*"In the current version of the manuscript, some severe limitations exist that call into question the usefulness of the kinetic parameters."*

We thank the reviewer for his/her critical remarks, which have allowed us to address the perceived limitations and add clarity to the manuscript. We recognise that a number of details were not presented in the main text and we have now included those in the revised manuscript. We now also present a sensitivity study that clarifies specifically the link between the derived rate coefficients and the desorption lifetimes that clearly was not explained adequately in the initial draft, but hopefully now better describes the usefulness of the kinetic parameters obtained. Furthermore, we have also amended the conclusions and abstract clarifying that the rate coefficients should be quoted together with the desorption lifetimes obtained during the fitting. We have also toned down our interpretation of the data on stearic acid to reflect the valid concerns voiced by the reviewer and have moved Fig. 10 to the ESI. We have also included more references to previous work to clearly identify what are established methods that have been used successfully in many previous studies and have added more detail on all of the areas that were flagged for attention by the reviewer.

*"I hope that in addressing the points below the authors can provide a more compelling description of their work and the validity of their conclusions."*

We outline below how we have addressed the concerns. We have added detail to the description of our experimental and modelling work including a new scheme, a new figure, a new table, and additional references to relevant literature. We have re-written our conclusions reflecting the additional material presented and have explained more clearly the validity of the kinetic parameters presented.

*"Main points:*
*1) The methodology associated with preparing the film should be detailed in the main text."*

Further details have been added to describe how we prepared the films. While neutron reflectometry is a sophisticated technique that cannot simply be found in a laboratory, there is in fact considerable precedent for its use in the study of amphiphilic layers at the air/water interface (see the historical and recent reviews by Lu et al. 2000, Narayanan et al. 2017 and Braun et al. 2017, for example). Furthermore, a considerable body of recent work is forming on the application of the technique to resolve the reaction kinetics of systems of atmospheric relevance (e.g. King et al. 2009; Thompson et al. 2010; King et al.

2010; Pfrang et al. 2014; Sebastiani et al. 2015; Skoda et al. 2017). We have clarified this point in the revised manuscript by ensuring that all of the references above are cited; we added details on the volumes and concentrations of the spreading solutions used.

New text in revised manuscript:
- p. 4, lines 23-25: "The solutions of organic molecules in chloroform were prepared shortly before the experiments and the concentrations are given in mg of solute in volume of solution: for $d_{34}$OA 1.41 mg ml$^{-1}$, for $d_{14}$POA 1.26 mg ml$^{-1}$, for $d_{33}$MO 1.11 mg ml$^{-1}$ and for $d_{35}$SA 0.58 mg ml$^{-1}$."
- p. 5, line 32 to p. 6, line 3: "Only a brief description of the physical basis of NR with reference to its application is given while more details may be found in Lu et al. (2000), Narayanan et al. (2017) and Braun et al. (2017). NR is a technique that can be used to measure the surface excess of oil-like films at the air–water interface. The scattering of neutrons is related to the coherent cross sections of the atoms with which they interact, and these values vary non-monotonically with respect to different isotopes of the same atom and different atoms across the periodic table. In particular, swapping hydrogen for deuterium in molecules changes significantly the scattering, and as such mixing of hydrogenous and deuterated materials enables contrast matching."
- p. 6, line 27-29: "A given amount of solution was spread using a microlitre syringe in order to form the monolayer following the protocol use in other NR studies of atmospheric relevance `(Pfrang et al. 2014; Sebastiani et al. 2015; Skoda et al. 2017; King et al. 2010; Thompson et al. 2010; King et al. 2009). The volume of solution spread was 24 µl for $d_{34}$OA, 23 µl for $d_{14}$POA, 32 µl for $d_{33}$MO and 35 µl for $d_{35}$SA."
- p. 6, line 31 to p. 7, line 8: "The trough in the reaction chamber did not have barriers to compress the film and adjust the surface pressure, hence the desired surface pressure, in the range of 16 to 25 mN m$^{-1}$ depending on the molecule, was achieved by spreading a calculated number of molecules on the water surface. Off-line tests using a surface pressure sensor confirmed that the surface pressure could be achieved reproducibly – between 2 to 7 % variation depending on the molecule – and the stability of the assembled film was assessed for 3–4 hours by monitoring the surface pressure or the reflectivity profile. From the surface excess obtained by NR the reproducibility is found to be within 1 to 9 %, depending on the molecule. The choice of initial surface pressure and surface excess was based on the requirement of maximising the signal-to-noise ratio for NR measurements while having a reaction that lasts long enough to be analysed for kinetics parameters. A reduction of the initial surface pressure is not expected to affect the kinetic behaviour, i.e. the $\Gamma(t)$ will start from a lower value and the curve will extend on a shorter time and less data will be available for the kinetic fitting. An increase of spread molecules will produce more droplets floating on top of a monolayer, when the molecule is unsaturated (compare to Figures 1–3 in Section 1 of ESI), while it will introduce inhomogeneity in the monolayer formed by saturated molecule (see Fig. 4 Section 1 of ESI) preventing a reliable interpretation of the NR measurement. The monolayer was further characterised with compression-expansion isotherms with a Langmuir trough off-line, while recording Brewster-angle microscopy (BAM) images at different surface pressure values, and these results are shown in the ESI Section 1."

"2) As far as I can tell, the compression of the film for the oxidation experiments is not reported anywhere. Please amend or make this information more prominent in the main text."

The experiments reported in the literature concerning the application of NR in studies of reaction kinetics of atmospheric relevance (see above) were performed typically using a large commercial Langmuir trough with barriers to control the surface pressure. Unfortunately this approach had the severe limitation that the trough was placed in a reaction chamber with a huge gas volume (ca. 50 L), which limited severely the range of oxidant concentrations that could be used as a result of the relatively large time for gas mixing.

For the kinetic experiments reported in the current study, we used our miniature reaction chamber (MIMIK chamber, described in Sebastiani et al., RSC Adv., 2015), which we developed purposely for this extensive study on a range of different systems. The technical challenge to develop an optimized chamber that has such a low gas volume (ca. 1 L) and that is compatible with NR and ellipsometry measurements should not be underestimated. By necessity, however, the chamber contained a small custom-made PTFE trough without barriers or pressure sensor, so it was not possible to measure the surface pressure in situ using this equipment.

As such, it was necessary for us to ensure that we could reproducibly spread the same amount of the starting material on the liquid surface for each system. Therefore we undertook offline calibration tests not only of the surface pressure (Wilhelmy plate), but also the surface excess (ellipsometry) and lateral morphology (Brewster angle microscopy) for all of the studied systems. We determined offline that we were able to reproducibly spread the same amounts within an error of 2–7 %. Further, it can be observed from the starting surface excess values measured directly using NR for each system that this reproducibility is in the range 1–9 %. Hence the need for barrier control in this work was convincingly circumvented, while a broad range of oxidant concentrations was accessed in a comprehensive study on different systems for the first time.

New text in revised manuscript:
-   p. 6, line 31 to p. 7, line 8: "The trough in the reaction chamber did not have barriers to compress the film and adjust the surface pressure, hence the desired surface pressure, in the range of 16 to 25 mN m$^{-1}$ depending on the molecule, was achieved by spreading a calculated number of molecules on the water surface. Off-line tests using a surface pressure sensor confirmed that the surface pressure could be achieved reproducibly – between 2 to 7 % variation depending on the molecule – and the stability of the assembled film was assessed for 3–4 hours by monitoring the surface pressure or the reflectivity profile. From the surface excess obtained by NR the reproducibility is found to be within 1 to 9 %, depending on the molecule. The choice of initial surface pressure and surface excess was based on the requirement of maximising the signal-to-noise ratio for NR measurements while having a reaction that lasts long enough to be analysed for kinetics parameters. A reduction of the initial surface pressure is not expected to affect the kinetic behaviour, i.e. the $\Gamma(t)$ will start from a lower value and the curve will extend on a shorter time and less data will be available for the kinetic fitting. An increase of spread molecules will produce more droplets floating on top of a monolayer, when the molecule is unsaturated (compare to Figures 1–3 in Section 1 of ESI), while it will introduce inhomogeneity in the monolayer formed by saturated molecule (see Fig. 4 Section 1 of ESI) preventing a reliable interpretation of the NR measurement. The monolayer was further characterised with compression-expansion isotherms with a Langmuir trough off-line, while recording Brewster-angle microscopy (BAM) images at different surface pressure values, and these results are shown in the ESI Section 1."

**"3) A more basic introduction to the NR technique is necessary, focusing on the observed quantities and what this actually means in these experiments. Where does the value of 'd' come from? Does 'd' change during an experiment in which shorter chain surface active species may be created, and how is this accounted for?"**

In our original submission we chose to reference the relevant literature since we did not think a greater level of detail would be particularly useful for the cross-disciplinary audience of *Atmospheric Chemistry and Physics*. Nevertheless, we take on board the suggestion of the reviewer and now provide a more detailed

description for non-experts in the technique and have added additional references where more details can be found.

The surface excess, $\Gamma = \frac{1}{A_{hg}} = \frac{\rho d}{b}$, was defined in the original manuscript, as was $d$ as the thickness of the layer in Å (p. 6, line 11-13), but the reviewer is correct to question if in fact $d$ was being measured explicitly or not. The answer is that as we were using the technique deliberately in a low $q$-range with a subphase that was matched in isotopic contrast to air, the measured neutron reflectivity was strictly related to the surface excess, which is proportional to the product of the density and thickness of the layer. Therefore in our model, it did not matter whether we fitted the density at fixed thickness or the thickness at fixed density. Only the product, which is proportional to the surface excess, mattered, and as such we did not have sensitivity to $d$ explicitly. We have clarified this point as well in the revised text.

New text in revised manuscript:
- p. 5, lines 27-31: "NR measurements of the oxidation of deuterated monolayers by $NO_3$ in the reaction chamber (Sebastiani et al., 2015) were carried out on FIGARO at the Institut Laue-Langevin (Campbell et al., 2011). High flux settings were used to maximise the data acquisition rate involving an incident angle, $\vartheta$, of 0.62°, a wavelength, $\lambda$, range of $2 - 20$ Å, and a constant resolution in momentum transfer, $q$, of 11% over the probed $q$-range of 0.007 to 0.07 Å$^{-1}$, where $q = 4\pi \sin \vartheta / \lambda$."
- p. 5, line 32 to p. 6, line 3: "Only a brief description of the physical basis of NR with reference to its application is given while more details may be found in Lu et al. (2000), Narayanan et al. (2017) and Braun et al. (2017). NR is a technique that can be used to measure the surface excess of oil-like films at the air–water interface. The scattering of neutrons is related to the coherent cross sections of the atoms with which they interact, and these values vary non-monotonically with respect to different isotopes of the same atom and different atoms across the periodic table. In particular, swapping hydrogen for deuterium in molecules changes significantly the scattering, and as such mixing of hydrogenous and deuterated materials enables contrast matching."
- p. 6, lines 13-23: "The surface excess for insoluble molecules corresponds to the surface concentration. A stratified layer model was applied to the experimental data involving a single layer for the deuterated surfactant. It has been shown that in such a case and in this low $q$-range ($< 0.07$ Å$^{-1}$), the value of $\Gamma$ is very insensitive to specific details of the model applied (Angus-Smyth et al. 2012). Therefore, fitting of the thickness with an arbitrary fixed value of the density or fitting of the density with an arbitrary fixed value of the thickness (each within reasonable bounds) gives equivalent results to within an added uncertainty of $< 2$ %. That is, only the fitted product $\rho d$ directly determines $\Gamma$, and the measurement approach deliberately desensitizes the data to structural information such as the actual layer thickness during the reaction in order to gain the requisite kinetic resolution. In our case, we chose to fit $\rho$ while fixing $d$ at the value obtained by fitting data recorded over a wider $q$-range (up to 0.25 Å$^{-1}$)."

***"4) Uptake of N2O5 into aqueous phases can lead to acidification as HNO3 is formed. How does the pH of the aqueous sub-phase change during the measurements, and how might changes in pH affect the film properties?"***
The pH was not monitored as in previous work we had varied the subphase composition and did not find strong effects; pH variation in King et al. (PCCP, 2009) did not show an effect on the reaction kinetics.

New text in revised manuscript:
- p. 21, lines 3-7: "King et al. (2009) investigated OA oxidation by $O_3$ on different subphases with pH ranging from 2 to 7 and no significant change was found in the rate coefficient. In our experiments with the oxidant $NO_3$ we expect $HNO_3$ to be formed, and induce a change in pH

in the subphase, but given the fact that it was previously reported that there was no pH effect, we did not explore the pH changes in the present study."

**"*Could changes in pH contribute towards the plateau observed in the initial time during some measurements? Might there be competition between uptake of N2O5 and NO3?*"**

Since the gas species are flowing constantly, it is not likely that the plateau is due to pH change caused by acidification of the subphase, since this will continue along the decay and no other effects are observed; the plateau is in our view more likely to be related to the film composition (monolayer & droplets/lenses). Competing adsorption of $N_2O_5$ and $NO_3$ to the monolayer can occur, however [$N_2O_5$] remains effectively constant for the gas-phase conditions we chose for varying [$NO_3$] and the clear variations in the decay behaviour observed indicate that the decay is dominated by $NO_3$-initiated oxidation of the monolayer. A number of studies have investigated reactive uptake of $N_2O_5$ and $NO_3$ confirming substantially faster uptake of $NO_3$ compared to $N_2O_5$ and a single study comparing uptake for oleic acid found a faster uptake by four orders of magnitude of $NO_3$. Further references and explanatory comments have been added to the manuscript. The evolution of the concentrations of the various gas-phase species is now explained in more detailed in the manuscript including a new Table 1 (in addition to the information that was presented in the ESI).

New text in revised manuscript:
-   p. 7, line 34 to p. 8, line 7: "Because of the method used to produce $NO_3$ (see Section 2.1.2 and Sections 2–3 of ESI) the ratio [$NO_2$]/[$NO_3$] increases from $10^5$ to $10^7$ as [$NO_3$] decreases from $10^9$ to $10^8$ molecule $cm^{-3}$. Since $NO_2$ can adsorb and desorb from the organic layer (compare King et al., 2010), occupying reactive sites for an average time represented by the desorption lifetime, the loss of organic material due to reaction with $NO_3$ may also be affected. The $NO_2$ occupies a reactive site, which becomes unavailable for $NO_3$ oxidation, and hence reduces the number of reactive sites available and slows down the apparent reaction rate. In particular, for high [$NO_2$]/[$NO_3$] ratios the reactant loss rate will be lower than the loss rate recorded for the lower [$NO_2$]/[$NO_3$] ratios. To take this effect into account we included the absorption and desorption of $NO_2$ in the model and to describe it we introduced the parameter called desorption lifetime, $\tau_{d,NO2}$, following the approach used by Shiraiwa et al. (2009). The effect of $N_2O_5$ is not considered in the model, since the concentration was constant for all gas conditions, as shown in Figure 8 of the ESI. Experimental studies of reactive uptakes of $NO_3$ and $N_2O_5$ (Gross & Bertram, 2008; Zhang et al., 2014a; Gržinic, et al., 2015) have shown that $NO_3$ uptake is substantially faster with a comparative study for OA reporting a ca. four orders of magnitude higher uptake coefficient of $NO_3$ compared to $N_2O_5$ (Gross et al., 2009)."
-   p. 5, lines 10-25: "At a total flow rate of l.2 to 1.5 $dm^3$ $min^{-1}$, [$NO_3$] ranged from $(3.5 \pm 1.5) \times 10^8$ ($13 \pm 5$ ppt) to $(2.3 \pm 1.2) \times 10^9$ molecule $cm^{-3}$ ($86 \pm 45$ ppt) in the experiments presented here; [$NO_3$] and $NO_2$ flow rates are given in Table 1. From the gas reaction model it is found that $NO_2$ reaches the steady state concentration faster when initial [$NO_2$] is higher. Ozone is consumed quantitatively in less than 250 s (see Fig. 7 in Section 3.1 of ESI). The concentration of $NO_3$ is lower the higher the excess of $NO_2$ (see Fig. 9 in Section 3.1 of ESI). The steady state concentrations of $N_2O_5$ are always approaching a similar value (see Fig. 8 in Section 3.1 of ESI) that is determined by the initial ozone concentration.

**Table 1.** The concentrations of $NO_3$ calculated from IR measurements of $[NO_2]$ and $[N_2O_5]$ are reported in the first column as molecule $cm^{-3}$ and the corresponding ppt value is given in the second column; in the third column the flow rate of $NO_2$ is shown (the total gas mixture flow rate is obtained by adding the constant $O_2$ flow rate of 1.2 $dm^3\,min^{-1}$ to these values).

| $NO_3$ / molecule $cm^{-3}$ | $NO_3$ / ppt | $NO_2$ flow rate / $dm^3\,min^{-1}$ |
|---|---|---|
| $(3.5 \pm 1.5) \times 10^8$ | $(13 \pm 5)$ | 0.360 |
| $(4.2 \pm 1.4) \times 10^8$ | $(15 \pm 5)$ | 0.290 |
| $(6.1 \pm 1.2) \times 10^8$ | $(23 \pm 4)$ | 0.200 |
| $(9 \pm 3) \times 10^8$ | $(32 \pm 10)$ | 0.160 |
| $(10 \pm 3) \times 10^8$ | $(36 \pm 10)$ | 0.130 |
| $(9.3 \pm 2.4) \times 10^8$ | $(35 \pm 9)$ | 0.104 |
| $(2.3 \pm 1.2) \times 10^9$ | $(86 \pm 45)$ | 0.08 |

The modelled concentrations were confirmed by IR measurements of $[NO_2]$ and $[N_2O_5]$ (the full dataset is displayed in Section 3.2 of ESI)."

**"5) *It is not clear what assumptions are made in order to derive the rate constants. In particular, the authors should perform a sensitivity analysis to see how changes in branching ratios affect the results.*"**

This is an important point raised by the reviewer and we thank him/her for the suggestion. We have added further detail on the assumptions made in our modelling and how we have assessed the sensitivity of our modelling approach to the key parameters. The branching ratios are based on literature values, as is detailed in the revised text. We have added an additional figure illustrating a sensitivity study of the impact of varying the desorption lifetimes. We have re-written conclusions and abstract to clarify the relation of the rate coefficients obtained to the fitted desorption lifetimes.

New text in revised manuscript:
- p. 7, lines 28-34: "In the model, the branching ratios for volatile and soluble products are based on literature values, and for surface-active products an estimation was based on $\Gamma(t)$ at long reaction times; the technique used in this study monitors the deuterium concentration at the interface, no other information can be obtained. We could have described the reaction system by assuming only two types of products: surface active and non-surface active. However, we decided to distinguish non-surface active compounds between volatile and soluble products in order to make our model suitable for description of experimental data probing the partitioning to subphase and/or gas-phase."
- p. 10, lines 6-9: "The product branching ratios affect the whole $\Gamma(t)$, varying $c_S$ the final value of $\Gamma(t)$ changes, i.e. a higher $c_S$ leads to a higher final value of $\Gamma(t)$; the model is less sensitive to changes in $c_G$ and $c_B$, however change in the solubilisation and/or volatilisation kinetic parameters ($D_{b,B}$ and $k_{loss,G}$) will affect the decay of $\Gamma(t)$. These parameters were chosen in order to best describe the experimental data and taking into account literature data."

**"6) *Looking at the data for OA vs POA, the uptake coefficients are similar, but the time constants are a factor of 2 different. What causes this?*"**

We now discuss in more detail the relation between uptake and desorption lifetimes (we assume this is meant by 'time constants' above) and have also amended conclusions and abstract to reflect this point.

New text in revised manuscript:
- p. 17, lines 10-16: "$\tau_{d,NO3,1}$ is double of the value found for oleic acid, this lifetime refers to the monolayer when is highly packed (see description in Section 2.2) and that is the condition where the difference in chain length between $d_{14}POA$ and $d_{34}OA$ can play a role. The higher value of $\tau_{d,NO3,1}$ for $d_{14}POA$ is consistent with the hypothesis of an easier access to the double

bond due to the shorter alkyl chain of $d_{14}$POA. The $\tau_{d,NO3,2}$ does not show a big difference between $d_{14}$POA and $d_{34}$OA and that refers to the monolayer in a less dense state, suggesting that once the access to the double bond is comparable the reaction has a similar behaviour for the two molecules."

- Abstract and p. 25, from line 37: "[NR experiments together with tailored kinetic modelling allowed us to determine the rate coefficients for the oxidation of OA, POA and MO monolayers to be $(2.8 \pm 0.7) \times 10^{-8}$ cm$^2$ molecule$^{-1}$ s$^{-1}$, $(2.4 \pm 0.5) \times 10^{-8}$ cm$^2$ molecule$^{-1}$ s$^{-1}$ and $(3.3 \pm 0.6) \times 10^{-8}$ cm$^2$ molecule$^{-1}$ s$^{-1}$,] for fitted initial desorption lifetimes of NO$_3$ at the closely packed organic monolayers, $\tau_{d,NO3,1}$, of $8.1 \pm 4.0$ ns, $16 \pm 4.0$ ns and $8.1 \pm 3.0$ ns, respectively. The approximately doubled desorption lifetime found in the best fit for POA compared to OA & MO is consistent with a more accessible double bond associated with the shorter alkyl chain of POA facilitating initial NO$_3$ attack at the double bond in a closely packed monolayer."

**"There appears to be no relation to NO2/NO3 adsorption lifetimes and uptake, so what purpose do they serve in the model?"**

The uptake is derived from the rate coefficient, while the desorption lifetime is an independent parameter describing the mean residence time of a gas molecule on a reactive site. Further details on the meaning of the kinetic parameters have been added in the manuscript and the relationship between rate coefficients and desorption lifetimes have been clarified throughout the manuscript including in the conclusions and abstract.

New text in revised manuscript:

- p. 9, lines 1-10: "In the case of NO$_2$ the corresponding equation 4 does not have the $L_{surf}$ term, since it is not reactive toward the organic molecules considered (King et al. 2010), Eq. 5 is the same. The flux of adsorbed gas molecules, $J_{ads, NO_3}$, is proportional to the surface accommodation coefficient, $\alpha_{s,NO3}$, is determined by the product of the surface accommodation coefficient on an adsorbate–free surface, $\alpha_{s,0,NO3}$, and the sorption layer coverage $\theta_s$ which is given by the sum of the surface coverage of all competing adsorbate species (see details in Section 4.1 of ESI). The flux of desorption, $J_{des, NO_3}$, is proportional to the inverse of the desorption lifetime, $\tau_{d,NO_3,eff}^{-1}$, which is the average time that the NO$_3$ molecule occupies an adsorption site. $\tau_{d,NO_3,eff}^{-1}$ is a combination of two desorption lifetimes, depending on the organic molecule packing at the interface, $\theta_{ss} = [Y]_{ss}(t)/[Y]_{ss}(0)$; either closely packed ($\tau_{d,NO_3,1}^{-1}$), or in the gas-like state ($\tau_{d,NO_3,2}^{-1}$):"

- p. 10, lines 10-28: "The kinetic model described above depends on several parameters, and some of them are strongly correlated. For example, for a given gas species time evolution, which may be described by certain accommodation coefficients ($\alpha_{s,0,Xi}$ where X$_i$ is NO$_3$ or NO$_2$) and certain desorption lifetimes ($\tau_{d, Xi}$), a good fit may be obtained as well with a lower $\alpha_{s,0,Xi}$ combined with a higher $\tau_{d,Xi}$. The accommodation coefficient represents the probability of the gas-phase molecule to absorb at the organic layer, hence the lower $\alpha_{s,0,NO3}$ is, the smaller is the probability of the reaction with the organic molecule. The desorption lifetime represents the mean residence time of the molecule absorbed at the surface, hence the longer this time, the higher is the probability for the gas molecule to react (valid for NO$_3$). NO$_2$ does not react with the organic layer (King et al. 2010), but those parameters still compensate, because $\alpha_{s,0,NO2}$ determines the number of molecules absorbed and $\tau_{d,NO2}$ determines the number of molecules leaving the sorption sites. The choice of leaving both of these parameters free to vary in the fitting will lead to a wide range of values for both. The resulting surface excess will match the experimental data. However, the choice of fixing one out of these two parameters makes the optimisation of the model computationally easier and the comparison between different organic molecules possible. In the fitting we have fixed the $\alpha_{s,0,Xi}$ to one for both gas species. The desorption lifetime for the reactive species, NO$_3$, shows a correlation to the reaction rate coefficient, $k_{surf,Y,NO_3}$, for example if the rate coefficient is kept constant an increase in desorption lifetime will lead to higher loss rate, and vice versa, if $\tau_{d, NO_3,eff}$ is kept constant and $k_{surf,Y,NO_3}$ increases the loss rate will augment. Our measurement follows the loss

rate, the values for $k_{\text{surf,Y,NO}_3}$ and $\tau_{\text{d, NO}_3\text{,eff}}$ are obtained from the best fit of the model to the data."

**"How would a change in adsorption lifetime manifest itself in the experimental data or the parameters they pull out?"**

An increase in desorption lifetime will increase the loss rate. We have added a detailed discussion and a sensitivity study varying the desorption lifetimes is presented as the new Figure 5.

New text in revised manuscript:

- p. 13, line 24 to p. 14, line 11: "Figure 5 displays a sensitivity study that demonstrates how the change of desorption lifetimes can affect the model while keeping all the other parameters to the best fit values. A decrease of $\tau_{\text{d, NO}_3\text{,2}}$ slows down the loss rate, especially for the second half of the decay, while an increase of $\tau_{\text{d, NO}_3\text{,1}}$ speeds up the decay substantially. A decrease in $\tau_{\text{d,NO2}}$ does not affect the model significantly ($\tau_{\text{d,NO2}}$ was reduced by four orders of magnitude to see any effect in Fig. 5), while an increase slows down the loss rate. Fig. 5 illustrates that the rate coefficients derived through modelling should be quoted together with the desorption lifetimes obtained for the best fit given the substantial impact of changes in the desorption times on the fit to the experimentally observed decays.

[Figure]

**Figure 5.** The experimental data for dOA exposed to [NO$_3$] = 86 ppt are shown with the best fit in red. The desorption lifetimes for NO$_2$ and NO$_2$ have selectively been modified in this sensitivity study to show their effect on the modelled surface excess decay. Condition 1 refers to $\left(\tau_{\text{d, NO}_3\text{,1}}\right)_{best\ fit} = \tau_{\text{d, NO}_3\text{,2}}$ , and hence $\tau_{\text{d, NO}_3\text{,2}} = \frac{1}{6}\left(\tau_{\text{d, NO}_3\text{,2}}\right)_{best\ fit}$ . Condition 2 refers to $\tau_{\text{d, NO}_3\text{,1}} = \left(\tau_{\text{d, NO}_3\text{,2}}\right)_{best\ fit}$ and hence $\tau_{\text{d, NO}_3\text{,1}} = 6\left(\tau_{\text{d, NO}_3\text{,1}}\right)_{best\ fit}$. Condition 3 refers to $\tau_{\text{d,NO2}}=10^{-4}$ $(\tau_{\text{d,NO2}})_{best\ fit}$. Condition 4 refers to $\tau_{\text{d,NO2}}=15$ $(\tau_{\text{d,NO2}})_{best\ fit}$."

**"7) What would the decay curves look like if all the products remained at the surface? Given that the technique can only provide information on the partitioning of products away from the surface, how can the authors be sure that multiple generations of oxidation are not occurring prior to material desorbing from the interface?"**

The surface excess would remain at a constant value and no decay curve would be observed if all of the reaction products were to remain at the interface. The technique is sensitive to the loss of material from the interface with time due to

solubilisation or volatilization of the products. It is well established that the primary reactive site for unsaturated fatty acids is the double bond; secondary reactions which we expect to be of relatively minor significance in our set-up with a one-molecule thin layer of organic molecules with a single reactive site in each molecule ($NO_3$-initiated hydrogen abstraction is much slower than addition to the double bond as demonstrated in our work on the saturated surfactant stearic acid) are not considered in our work and could not be resolved in this experimental approach. Therefore multiple generations of oxidation products are not considered explicitly in this work.

We are not aware of another technique than NR that could be used to follow the chemical composition of reactants versus products in thin films on a water subphase with suitable time resolution to resolve the kinetic behaviour of the reactions as we have managed in the present work. Simultaneous neutron reflectometry and infrared reflection absorption spectroscopy (IRRAS), a technique we have recently developed for study of related systems (Skoda et al., RSC Adv., 2017) may be able to give some information on the chemical composition in the future.

New text in revised manuscript:
- p. 22, lines 1-9: "For all the reactions studied here we expect secondary reactions not to be significant due to our set-up with a one-molecule thin layer of organic molecules each containing only a single reactive site ($NO_3$-initiated hydrogen abstraction is much slower than addition to the double bond as demonstrated in our work on the oxidation of the saturated surfactant stearic acid). Multiple generations of oxidation products could not be resolved in this experimental approach and are not considered explicitly in this work. Simultaneous neutron reflectometry and infrared reflection absorption spectroscopy (IRRAS), a technique we have recently developed for study of related systems (Skoda et al., 2017) may be able to give some information on the chemical composition of one-molecule thin films during kinetic studies of oxidation reactions at the air–water interface in the future."

**"8) *The modelling is performed in such as way that it is not clear if there is any predictive power to the results. For example, the model is fit to the initial decay, and then floated for the remaining time, and in most cases this free-floating region does not do a good job as describing the data. Is this because additional processes are occurring that are not factored into the model?"***

We recognise that the description of our modelling approach was not adequately clear in the initial submission and we have added further detail and analysis (see sensitivity study on the impact of varying the desorption lifetimes described above) in the revised manuscript. We have clarified the relation between rate coefficients and desorption lifetimes throughout the manuscript and believe that we have demonstrated that there is predictive power in our approach and clarified the reasons why we use the 'middle sections' of the decays for our fitting given the additional uncertainties associated with the initial and final sections of the decays as pointed out by the reviewer.

New text in revised manuscript:
- p. 12, lines 11-17: "The range of data used for the kinetic fitting starts after the initial plateau, and ends at $1 \times 10^{14}$ molecule $cm^{-2}$: data below this value are excluded from the fitting for two main reasons: (i) at low coverage the data become more sensitive to experimental details such as the precise background subtraction, so the parameters that affect the kinetic model are better determined without increasing sensitivity to these factors; and (ii) at low coverage some surfactants can segregate into domains which are inhomogeneous laterally, and the NR model

does not have the resolution to distinguish this effect but the results are modestly affected, so again it is better to desensitize the kinetic parameters to this effect."

- p. 13, lines 16-24: "From BAM images (see ESI) we know that droplets of organic molecules float on top of the monolayer, we need to account for this extra molecules when fitting the model to the experimental data. In fact, the molecules of the monolayer and the droplets are consumed upon oxidation but, until droplets are present, they act as a reservoir and further molecules from the droplets may spread and maintain a constant surface excess until the droplets disappear, leading to the delayed start in decay. The NR signal is averaged over a large surface ($cm^2$) and it is not sensitive to small droplets ($\mu m$) thicker than the monolayer, that is why the surface excess value is constant for this initial part of the decay. To account for this, the initial value for the theoretical $\Gamma(t)$ was adjusted to a higher value than the initial experimental plateau value and the experimental data were considered for fitting after the initial plateau ended (see Figure 5)."

- p. 10, lines 10-28 (see also above): "The kinetic model described above depends on several parameters, and some of them are strongly correlated. For example, for a given gas species time evolution, which may be described by certain accommodation coefficients ($\alpha_{s,0,Xi}$ where $X_i$ is $NO_3$ or $NO_2$) and certain desorption lifetimes ($\tau_{d,Xi}$), a good fit may be obtained as well with a lower $\alpha_{s,0,Xi}$ combined with a higher $\tau_{d,Xi}$. The accommodation coefficient represents the probability of the gas-phase molecule to absorb at the organic layer, hence the lower $\alpha_{s,0,NO3}$ is, the smaller is the probability of the reaction with the organic molecule. The desorption lifetime represents the mean residence time of the molecule absorbed at the surface, hence the longer this time, the higher is the probability for the gas molecule to react (valid for $NO_3$). $NO_2$ does not react with the organic layer (King et al. 2010), but those parameters still compensate, because $\alpha_{s,0,NO2}$ determines the number of molecules absorbed and $\tau_{d,NO2}$ determines the number of molecules leaving the sorption sites. The choice of leaving both of these parameters free to vary in the fitting will lead to a wide range of values for both. The resulting surface excess will match the experimental data. However, the choice of fixing one out of these two parameters makes the optimisation of the model computationally easier and the comparison between different organic molecules possible. In the fitting we have fixed the $\alpha_{s,0,Xi}$ to one for both gas species. The desorption lifetime for the reactive species, $NO_3$, shows a correlation to the reaction rate coefficient, $k_{surf,Y,NO_3}$, for example if the rate coefficient is kept constant an increase in desorption lifetime will lead to higher loss rate, and vice versa, if $\tau_{d,NO_3,eff}$ is kept constant and $k_{surf,Y,NO_3}$ increases the loss rate will augment. Our measurement follows the loss rate, the values for $k_{surf,Y,NO_3}$ and $\tau_{d,NO_3,eff}$ are obtained from the best fit of the model to the data."

- Abstract and p. 25, from line 37 (see also above): "[NR experiments together with tailored kinetic modelling allowed us to determine the rate coefficients for the oxidation of OA, POA and MO monolayers to be $(2.8 \pm 0.7) \times 10^{-8}$ $cm^2$ $molecule^{-1}$ $s^{-1}$, $(2.4 \pm 0.5) \times 10^{-8}$ $cm^2$ $molecule^{-1}$ $s^{-1}$ and $(3.3 \pm 0.6) \times 10^{-8}$ $cm^2$ $molecule^{-1}$ $s^{-1}$,] for fitted initial desorption lifetimes of $NO_3$ at the closely packed organic monolayers, $\tau_{d,NO3,1}$, of $8.1 \pm 4.0$ ns, $16 \pm 4.0$ ns and $8.1 \pm 3.0$ ns, respectively. The approximately doubled desorption lifetime found in the best fit for POA compared to OA & MO is consistent with a more accessible double bond associated with the shorter alkyl chain of POA facilitating initial $NO_3$ attack at the double bond in a closely packed monolayer."

*"Can the parameters obtained be used to accurately predict the chemistry at different film compressions, oxidant concentration etc.?"*

The best fit parameters can predict the fate of an organic monolayer with a different compression, i.e. a different initial surface excess, as well as different oxidant concentration. Further details of the model are added as described above. Furthermore we added details on the impact of changes in the surface excess and on observations made in complementary work using Brewster angle microscopy (BAM).

New text in revised manuscript:

- p. 22, lines 11-14: "Although our present approach did not allow convenient variation of the surface excess due to the barrier-less Langmuir trough in our miniature kinetic chamber optimised for kinetic measurements of fast reactions (Sebastiani et al., 2015), we believe that the best fit parameters we report in the present study can predict the fate of an organic monolayer with a different compression, i.e. at a different initial surface excess."

**"9) In the model, partitioning away from the surface (either into the gas phase or bulk aqueous solution) is rapid. Is there any consequence in the modelling for partitioning to one or the other?"**

The model cannot distinguish between removal of products to the gas phase or subphase; for the model the only important feature of the partitioning is the diffusion of material away from the interface. We have clarified this in the revised manuscript.

New text in revised manuscript:

- p. 7, lines 28-34: "In the model, the branching ratios for volatile and soluble products are based on literature values, and for surface-active products an estimation was based on $\Gamma(t)$ at long reaction times; the technique used in this study monitors the deuterium concentration at the interface, no other information can be obtained. We could have described the reaction system by assuming only two types of products: surface active and non-surface active. However, we decided to distinguish non-surface active compounds between volatile and soluble products in order to make our model suitable for description of experimental data probing the partitioning to subphase and/or gas-phase."

**"In these experiments, would the same results be obtained if the products were simply broken down into surface-present and surface-absent?"**

Yes, but we decide to add the further distinction to make the model more complete and suitable for describing experiments that can follow the products partitioning to the subphase/gas-phase. The text added to the revised manuscript is inserted above.

**"How are partially surface active molecules accounted for in the present analysis?"**

Assuming 'partially surface active' refers to molecules that would stay at the surface for a finite time (shorter than the experiment duration), we have no evidence from the modelling of the data of the existence of these kind of products: their presence would make the modelled decay deviate clearly towards a slower loss; if the lifetime of those species would be longer than the experiment duration they could not be distinguished from the surface active products. We average over a large area and are sensitive only to the net surface active products; if there would be quickly adsorbing-desorbing products we would be unable to detect these processes since we measure only a time and area average.

**"10) The data for stearic acid is not convincing – while it clearly shows less reactivity, the magnitude of the decay is very small. I would suggest this be removed or moved to the SI to allow for the additional material in the main text to address the previous points in this review."**

We have removed Figure 10 with the data for stearic acid and the kinetic fit and have added it in the ESI instead in line with the referee's comments. We have

also added additional material to address the points raised in this review specifically adding a new Scheme, a new Table and a new Figure. We have also toned down the interpretation of our experimental findings for stearic acid.

New text in revised manuscript:
- p. 20, lines 2-4: "the model fit to the experimental data is displayed in the ESI; the estimated kinetic parameters should be considered with caution given the severe limitations mainly due to the lack of experimental data."
- abstract: "For the much slower $NO_3$-initiated oxidation of the saturated surfactant SA we estimated a loss rate of approximately $(5 \pm 1) \times 10^{-12}$ cm$^2$ molecule$^{-1}$ s$^{-1}$ which we consider to be an upper limit for the reactive loss, and estimated an uptake coefficient of ca. $(5 \pm 1) \times 10^{-7}$"

*"Minor points:*
  *1) A figure in the main text showing the structure of the molecules would make comparisons of the datasets easier for a reader."*
The structures of the organic molecules studied are now included in the revised manuscript.

New text in revised manuscript:
- p. 4, lines 17-18: "the chemical structures of the molecules studied are displayed in Scheme 1."

stearic acid

methyl oleate

oleic acid

palmitoleic acid

**Scheme 1.** Chemical structures of the organic molecules studied.

*"2) The reference to Section 4 of the SI containing examples of raw data is wrong – there is no raw data presented in the SI".*
The reference to 'raw data' has been removed.

*"3) Consistency of units (some mixing of [m] and [cm] between text and figures when reporting surface excess."*
This has been corrected in the revised manuscript.

*"4) Define acronyms consistently, even if they are well known in your field (e.g. FIGARO, PRA)"*
This has been corrected in the revised manuscript and we have made sure all acronyms are defined consistently.

*"5) Define the term "surface excess" – it is unclear in the derivation if this is actually a surface concentration, as equation 2 seems to indicate. If I am correct in my understanding of the difference, for very insoluble species surface concentration and surface excess are approximately equal, but please clarify this."*

We have clarified this in the revised manuscript.

New text in revised manuscript:
- p. 6, lines 13-14: "The surface excess for insoluble molecules corresponds to the surface concentration."

*"6) The first paragraph of the "Discussion" would be more appropriate in either the Introduction or the Conclusions."*

The first paragraph of the discussion section has been slightly amended and been moved to the conclusions.

New text in revised manuscript:
- p. 25, lines 18-33: "We have investigated the reactions of the key atmospheric oxidant $NO_3$ with organic monolayers at the air–water interface as proxies for the night-time ageing of organic-coated aqueous aerosols. The surfactant molecules chosen allowed the investigation of the effects of chain length, head group properties and degree of unsaturation on the reaction kinetics as well as the proportion of surface-active products formed. The experimental results presented together with the tailored modelling approach for the four structurally different monolayers has allowed determination of the kinetic parameters of heterogeneous reactions at the air–water interface with $NO_3$ for the first time. The study of heterogeneous reactions of organic monolayers at the air–water interface exposed to oxidants is crucial to understand the role of such films for the atmospheric fate of organic-coated aqueous aerosols (Gilman et al., 2004). Previous studies performed on these types of reactions were nearly exclusively carried out monitoring the gas-phase species (Wadia et al., 2000; Knopf et al., 2007; Cosman et al., 2008a; Cosman et al., 2008b). Gross & Bertram (2009) investigated the oxidation of organic monolayers at an air–solid interface and in addition to monitoring the gas-phase species during the reaction, they analysed the product film with several surface spectroscopic techniques. The monitoring of the organic monolayer during oxidation at the air–water interface was introduced by King et al. (2009) for the study of OA exposed to $O_3$. To the best of our knowledge, no-one has previously investigated the oxidation of organic monolayer at the air–water interface by $NO_3$ by in situ kinetic measurements of the surface excess. "

*"7) Slightly excessive 'keywording'"*

We have removed the three keywords 'kinetics', 'oxidation' and 'air-water interface' from the revised manuscript.

**Referee #2**

*Major*

*"The authors give the relevance of the manuscript towards insight into organic coated aerosol (first sentence in the abstract). However, the experimental data is not taken from aerosol but from a flat surface, which could have a link to the marine boundary layer. This is significant given the known deviation of aerosols from the behaviour expected from a flat surface, that is more pronounced as the droplet size decreases and surface curvature increases. I would expect there to be a significant effect due to change in surface tension that would alter the kinetics between aerosol and flat surfaces that makes a flat surface an unsuitable experimental model for aerosol."*

Flat surfaces have been used as proxy for aerosol in many previous studies published in the literature. Curvature effects are unlikely to be significant for the processes studied here as the macroscopic size even of small droplets is far greater that the molecular length scales. It is true that transport rates in the bulk can be affected as a result of droplet curvature, but our system involves reactions of a gas-phase radical ($NO_3$) at the air–water interface where liquid diffusion is not a limiting factor. We have confirmed this previously in studies of various sized droplets reported elsewhere, and in these cases the flat monolayer allows unique insight into the reaction kinetics of one-molecule thin films. Indeed we have demonstrated in the present work that the recent advances in neutron reflectometry instrumentation have allowed us to access the kinetic information reported for the first time. It is true that we needed a model platform in order to access the information reported, but as the technique cannot be applied to droplets, we believe that our approach is fully justifiable. It follows significant precedent (e.g. King et al., PCCP, 2009, Pfrang et al., PCCP, 2014, Sebastiani et al., RSC Adv, 2015, Skoda et al., RSC Adv, 2017, Thompson et al., Langmuir, 2010 and King et al., Atmos Environ, 2010), yet the challenging technical advances in sample environment have in fact allowed us to perform a comprehensive study on systems not previously accessible. Curvature effects on the surface tension are –of course– not directly accessible by the experimental studies of one-molecule thin films on a flat water subphase presented here, but the PRA modelling framework we used in the present study is fully able to describe curved surfaces as we have done in the past (Pfrang et al., ACP, 2010; Pfrang et al., ACP, 2011; Shiraiwa et al., ACP, 2010 and Shiraiwa et al., ACP, 2012), so that the parameters we obtained in the PRA model variant presented here which we created specifically to describe a flat surface could directly be used e.g. in the KM-GAP model variant (Shiraiwa et al., ACP, 2012) to establish any curvature effects; however, we would not expect any large or particularly interesting effects for the processes studied here. However, we have used this opportunity to clarify our approach to the readers with the added text detailed below.

Please note that we are not aware of any experimental approach that could follow the kinetics of one-molecule thin films on curved surfaces with time resolutions required for the reactions studied here; this to our knowledge the first study of kinetics of one-molecule thin films floating on a water surface in reaction with nitrate radicals.

New text in revised manuscript:
- p. 8, lines 10-12: "Different to the model presented by Shiraiwa et al. (2010) we had to remove the curvature terms from the modelling code to be able to describe the flat air–water interface present in our experimental system. We do not expect any significant impact of curvature on the processes studied here."

***"Although the authors do note on page 21 that they are currently looking at aerosol proxies, none of this data appears to be included in the manuscript."***
These are entirely separate studies on ultrasonically levitated droplets probed simultaneously by Raman spectroscopy and synchrotron small angle X-ray scattering (SAXS) recently published in *Journal of Physical Chemistry Letters* (Seddon et al., 2016) with the latest work in press with *Nature Communications* (Pfrang et al., 2017). References to this work on complex 3D self-assembly have been added to the manuscript.

***"The experimental system has been previously outlined in Sebastiani, RSC Adv (2015), and is highly relevant to this manuscript. But even so, there does need to be more in-depth discussion of the experimental techniques used here particularly where these form the basis for drawing conclusion or inference. As the manuscript stands very little detail is given on the experimental setup and measurement, and crucially the conditions that measurements are taken under."***
The reviewer is correct to point out that an in-depth discussion of the experimental technique was not included in the original manuscript. We did include relevant citations and had the doubt that a more detailed discussion may not be ideally suited for the general readership of *Atmospheric Chemistry and Physics*. However, we have now used the opportunity to clarify this point considerably. Also, more detail has been added in the revised manuscript concerning monolayer preparation and preliminary characterisation of the monolayer, gas phase reaction described and table with full details on $NO_3$ concentration and flow rates (Section 2.1). An additional introduction on NR has been added. For example:

New text in revised manuscript:
- p. 4, lines 23-25: "The solutions of organic molecules in chloroform were prepared shortly before the experiments and the concentrations are given in mg of solute in volume of solution: for $d_{34}$OA 1.41 mg ml$^{-1}$, for $d_{14}$POA 1.26 mg ml$^{-1}$, for $d_{33}$MO 1.11 mg ml$^{-1}$ and for $d_{35}$SA 0.58 mg ml$^{-1}$."
- p. 5, lines 27-31: "NR measurements of the oxidation of deuterated monolayers by $NO_3$ in the reaction chamber (Sebastiani et al., 2015) were carried out on FIGARO at the Institut Laue-Langevin (Campbell et al., 2011). High flux settings were used to maximise the data acquisition rate involving an incident angle, $\vartheta$, of 0.62°, a wavelength, $\lambda$, range of $2 - 20$ Å, and a constant resolution in momentum transfer, $q$, of 11% over the probed $q$-range of 0.007 to 0.07 Å$^{-1}$, where $q = 4\pi \sin \vartheta / \lambda$."
- p. 5, line 32 to p. 6, line 3: "Only a brief description of the physical basis of NR with reference to its application is given while more details may be found in Lu et al. (2000), Narayanan et al. (2017) and Braun et al. (2017). NR is a technique that can be used to measure the surface excess of oil-like films at the air–water interface. The scattering of neutrons is related to the coherent cross sections of the atoms with which they interact, and these values vary non-monotonically with respect to different isotopes of the same atom and different atoms across the periodic table. In particular, swapping hydrogen for deuterium in molecules changes significantly the scattering, and as such mixing of hydrogenous and deuterated materials enables contrast matching."

- p. 6, lines 13-23: "The surface excess for insoluble molecules corresponds to the surface concentration. A stratified layer model was applied to the experimental data involving a single layer for the deuterated surfactant. It has been shown that in such a case and in this low $q$-range (< 0.07 Å$^{-1}$), the value of $\Gamma$ is very insensitive to specific details of the model applied (Angus-Smyth et al. 2012). Therefore, fitting of the thickness with an arbitrary fixed value of the density or fitting of the density with an arbitrary fixed value of the thickness (each within reasonable bounds) gives equivalent results to within an added uncertainty of < 2 %. That is, only the fitted product $\rho d$ directly determines $\Gamma$, and the measurement approach deliberately desensitizes the data to structural information such as the actual layer thickness during the reaction in order to gain the requisite kinetic resolution. In our case, we chose to fit $\rho$ while fixing $d$ at the value obtained by fitting data recorded over a wider $q$-range (up to 0.25 Å$^{-1}$)."
- p. 6, line 27-29: "A given amount of solution was spread using a microlitre syringe in order to form the monolayer following the protocol use in other NR studies of atmospheric relevance `(Pfrang et al. 2014; Sebastiani et al. 2015; Skoda et al. 2017; King et al. 2010; Thompson et al. 2010; King et al. 2009). The volume of solution spread was 24 μl for $d_{34}$OA, 23 μl for $d_{14}$POA, 32 μl for $d_{33}$MO and 35 μl for $d_{35}$SA."
- p. 6, line 31 to p. 7, line 8: "The trough in the reaction chamber did not have barriers to compress the film and adjust the surface pressure, hence the desired surface pressure, in the range of 16 to 25 mN m$^{-1}$ depending on the molecule, was achieved by spreading a calculated number of molecules on the water surface. Off-line tests using a surface pressure sensor confirmed that the surface pressure could be achieved reproducibly – between 2 to 7 % variation depending on the molecule – and the stability of the assembled film was assessed for 3–4 hours by monitoring the surface pressure or the reflectivity profile. From the surface excess obtained by NR the reproducibility is found to be within 1 to 9 %, depending on the molecule. The choice of initial surface pressure and surface excess was based on the requirement of maximising the signal-to-noise ratio for NR measurements while having a reaction that lasts long enough to be analysed for kinetics parameters. A reduction of the initial surface pressure is not expected to affect the kinetic behaviour, i.e. the $\Gamma(t)$ will start from a lower value and the curve will extend on a shorter time and less data will be available for the kinetic fitting. An increase of spread molecules will produce more droplets floating on top of a monolayer, when the molecule is unsaturated (compare to Figures 1–3 in Section 1 of ESI), while it will introduce inhomogeneity in the monolayer formed by saturated molecule (see Fig. 4 Section 1 of ESI) preventing a reliable interpretation of the NR measurement. The monolayer was further characterised with compression-expansion isotherms with a Langmuir trough off-line, while recording Brewster-angle microscopy (BAM) images at different surface pressure values, and these results are shown in the ESI Section 1."

***"Has the effect of NO3 flow rate on oxidation been considered? From page 4 there seems to be a range used, but details are not given with each measurement."***

Yes, this is an important point raised by the reviewer, and it is explained in the revised manuscript. Details on the concentrations and flow rates used have been added as detailed in response to reviewer 1 on page 5 including a new Table 1.

***"On page 5 the authors mention the laser alignment window, which seems to be for a 632.8 nm Helium Neon laser mentioned in Sebastiani, RSC Adv (2015). Is this experimental setup appropriate for this measurement given the intention to measure night-time oxidation? There needs to be further discussion to establish the conditions that the samples were kept under, e.g. was a dark room used to prepare samples, or an indication is needed that this has been considered."***

We assume the referee is concerned that the alignment laser could lead to photolysis of the photolabile NO$_3$ which has particularly a strong absorption band at 662 nm. We can confirm that care has been taking that the experiments are carried out in the dark with the mixing bulb (where NO$_3$ is being generated *in-situ*) being carefully covered by aluminium foil during all experimental runs;

the reaction chamber is made from aluminium and the tubing from PTFE, so that exposure to external light is avoided; the alignment laser is essential for the experimental studies to ensure that the air–water interface remains aligned with the neutron beam position throughout each kinetic experiment. The alignment laser is a very weak laser with less than 1 mW max. power output at source; the laser light has to travel through a glass window before entering the gas-phase environment that contains $NO_3$; it should also be noted that the gas-phase volume with a steady state $NO_3$ concentration is ca. 1L with a spot size of the laser of only ca. 120 μm, so that we are confident that the laser does not contribute any significant $NO_3$ loss compared to the other uncertainties associated with [$NO_3$] (see new Table 1 for the uncertainties associated with [$NO_3$] based on our spectroscopic measurements and associated modelling of the gas-phase concentrations of the nitrogen oxides present). Nevertheless, we are grateful to the reviewer for raising this point, as other readers of our work may also be concerned by our approach. Therefore we have added the following new text to clarify our methodology.

New text in revised manuscript:
-   p. 5, lines 6-8: "we ensured that the reaction chamber as well as the reaction bulb where $NO_2$ was allowed to react with $O_3$ to form $NO_3$ was kept in the dark to avoid any photolysis of the photolabile $NO_3$ during the experiments."
-   p. 7, lines 10-11: "(LK-G152, Keyence, Japan; laser class II, wavelength 650 nm, power output 0.95 mW, spot diameter 120 μm),"

***"Further discussion on the potential products that could form should be included. Without compositional information the assumptions made during the oxidation process and kinetic analysis are not convincing. Have the authors assumed that the products will not undergo further oxidation or degradation?"***
The technique applied is sensitive to surface concentration of deuterated material at the air–water interface (where the subphase is matched in isotopic contrast to air effectively to make its contribution invisible). The rate of loss of material with time is measured, and it follows that the products that are lost from the interface are more soluble or volatile than the reactants. In fact our model does not presume the fate of the products. At the same time, it is known that OA, PA and MO have one highly reactive site, a double bond that is cleaved by oxidants. Secondary reactions such as hydrogen abstraction would be expected to be minor (as confirmed in our experiments with the saturated surfactant stearic acid), and therefore they are not considered explicitly in our model. We have taken the opportunity to clarify this point in the revised version.

New text in revised manuscript:
-   p. 22, lines 1-9: "For all the reactions studied here we expect secondary reactions not to be significant due to our set-up with a one-molecule thin layer of organic molecules each containing only a single reactive site ($NO_3$-initiated hydrogen abstraction is much slower than addition to the double bond as demonstrated in our work on the oxidation of the saturated surfactant stearic acid). Multiple generations of oxidation products could not be resolved in this experimental approach and are not considered explicitly in this work. Simultaneous neutron reflectometry and infrared reflection absorption spectroscopy (IRRAS), a technique we have recently developed for study of related systems (Skoda et al., 2017) may be able to give some information on the chemical composition of one-molecule thin films during kinetic studies of oxidation reactions at the air–water interface in the future."

*"Minor Figure 4 appears before table 1, but look to be discussed in a different order in the text."*
The position of Fig. 4 has been corrected.

*"Acronyms such as BAM should be written in full. Symbols used in equations should be explained for clarity in the text, e.g. in section 2.1.3; lambda and theta."*
We have defined all acronyms consistently.

*"Page 13: The variable parameters on line 13 should have "is", an equals sign, "varied between", or something appropriate between symbol and values."*
"varied between" has been added where appropriate.